# DSparsE: Dynamic Sparse Embedding for Knowledge Graph Completion Task

## Abstract

Addressing the incompleteness problem in knowledge graphs remains a significant challenge. Current graph completion methods, such as ComDensE (a representative of the fully connected network) and InteractE (a representative of the convolutional network), have certain limitations. Specifically, ComDensE is prone to overfitting and has constraints on network depth, while InteractE has limitations in feature interaction and interpretability. To overcome these drawbacks, we propose the Dynamic Sparse Embedding (DSparsE) model. This model employs sparse linear structure, replacing the conventional dense layers with adaptable sparse ones. By applying sparse structure to the dynamic MLP layers composed of multiple expert blocks and the residual MLP layers in decoders, DSparsE maintain the network's robustness while ensuring the capability for feature interaction. Furthermore, through dimensionality reduction and visualization of the feature space output by the gating layers, we discovered that the hypernetwork structure within the dynamic layers is capable of effectively modeling various semantic associations between entity-relation pairs. A comprehensive series of experiments on general datasets demonstrate that DSparsE achieves state-of-the-art performance across multiple metrics.

## 1 Introduction

Knowledge graph (KG) is a directed heterogeneous graph that represents concepts, entities, and their relationships in a structured form using knowledge triples. Knowledge triples are typically represented as $(s, r, o)$, where $s, r$ and $o$ denote the subject entity, the relation, and the object entity, respectively. KGs have a wide range of applications in various fields, including natural language processing, information retrieval, recommendation systems, and semantic web technologies. They are used to represent and organize knowledge in a structured and machine-readable format, which can be used to power intelligent applications and services.

Some well-known KGs, including Wikidata (Vrandecic & Krtoetzsch, 2014) and DBpedia (Auer et al., 2007) contain billions of knowledge triples, but they are often incomplete, which poses a significant challenge in the field of knowledge graph research. To address this issue, knowledge graph completion has emerged as an important task, which aims to predict missing knowledge triples. Link prediction, a subtask of knowledge graph completion, focuses on predicting the missing entity in a knowledge triple. Graph embedding, which uses low-dimensional, dense, and continuous vectors to represent nodes and relationships in knowledge graphs, is the basis of most link prediction methods. Existing link prediction models can be categorized into (Rossi et al., 2021) tensor decomposition models (Kolda & Bader, 2009), translational models, and deep learning models (Rossi et al., 2021). Recently, pre-trained language models, such as LLM (Large Language Model) in NLP, have also been introduced into KG completion tasks.

This paper focuses on deep learning models for link prediction, starting from simple fully connected models and convolutional neural network models, and optimizing their network structures by introducing dynamic and sparse layers. ComDensE retains both shared fully connected layers and relation-aware fully connected layers, and concatenates their results in the projection layer to achieve feature fusion. The relation-aware layer can be seen as a dynamic weight that changes with the input data. However, this dynamic processing is not comprehensive because the weights of the shared layers are still fixed, which limits the network's expressive power. MoE (Shazeer et al., 2017;

Jacobs et al., 1991) and CondConv (Yang et al., 2019) were proposed in 2017 and 2019, respectively. The former divides the fully connected layer into several expert layers and uses a separate network to generate the combination weights of these expert layers. It takes the expert blocks with the top $k$ weights for feature fusion. The latter uses dynamic convolution kernels based on input data for convolution operations. These dynamic methods give the network greater flexibility and have been shown to have good application potential. DSparsE introduces a structure similar to MoE into the encoding end, but it takes the results of all expert blocks for weighted fusion instead of taking the results of the top $k$ expert blocks. This method could improve performance without a significant increase of parameters.

Compared to fully connected networks, convolutional layers introduce position-related sparse connections, which effectively suppress overfitting, save computing resources, and efficiently capture feature correlations between adjacent pixels. However, in the task of knowledge graph link prediction, the input to the neural network is a one-dimensional embedding vector, which does not naturally have position information like pixels in images. Most of the aforementioned convolution-based models (Dettmers et al., 2018; Nguyen et al., 2017; Jiang et al., 2019; Vashishth et al., 2020) attempt to enhance the interaction between entity and relation embedding vectors in different dimensions through different method. These novel methods achieved good results on many datasets, but they still suffer from insufficient feature interaction and interpretability (Because the neighborhood information of a point does not have actual meaning). Therefore, this paper considers directly using sparse layers with adjustable sparsity to replace all dense layers. Sparse layers can be seen as an upgrade to convolutional layers, while at the same time alleviating the overfitting issues faced by dense layers through unstructured pruning (Dettmers & Zettlemoyer, 2019).

In addition, the research of ComDensE (Kim & Baek, 2022) shows that the effect of a single wide network layer is even better than that of a deep network. This paper introduced residual connections (He et al., 2016) to solve the problem of the inability to deepen network models. In summary, the contributions of this paper can be listed as follows:

- We propose a novel knowledge graph link prediction model DSparsE, which introduces a dynamic layer into the encoding end and a residual structure into the decoding end. This enables neural networks to better perform information fusion and has the potential to deepen the network's layers.

- By replacing the fully connected layer with a sparse layer, our model effectively mitigates overfitting risks, all the while preserving its capacity for feature interactions. Significantly, at comparable interaction levels, fixed sparse structures demonstrate enhanced predictability compared to methods like Dropout or the reduction of output dimensions.

- By applying dimensionality reduction to the output of the gating layer within the dynamic layer, we discovered that the gating structure distributes weights to expert blocks based on the semantic information of node-relation pairs. The visualization results demonstrate that the hypernetwork structure is capable of effectively modeling semantic inverse relationships, antonymic relationships, and similar relationships. Additionally, the entities within the entity-relation pairs induce slight shifts in the hypernetwork's output.

- Tests conducted on FB15k-237, WNRR18, and YAGO3-10 demonstrate that our proposed model achieves state-of-the-art performance across multiple metrics. A series of extensive ablation studies and comparative experiments further elucidate the interconnections and efficacy of different components within the model.

## 2 BACKGROUND

A knowledge graph is a collection of triples (facts) that represent relationships between entities, denoted as $\mathcal{G} = \{(s, r, o)\} \subseteq \mathcal{E} \times \mathcal{R} \times \mathcal{E}$, where $s \in \mathcal{E}$ and $o \in \mathcal{E}$ are the triple subject and object, respectively, and $r \in \mathcal{R}$ is the relationship between them. The link prediction task in KGs can be regard as a point-wise learning to rank problem, where the goal is to learn a scoring function that maps an input triple $t = (s, r, o)$ to a score $\psi(t)$ that reflects the likelihood of the fact encoded by $t$ being true, denoted as $\psi : \mathcal{E} \times \mathcal{R} \times \mathcal{E} \mapsto \mathbb{R}$. In other words, the objective is to predict the missing entity in a triple given the other two entities and the relationship between them using a mapping function. **Due to space constraints, we have placed the related work in Appendix A, while some typical scoring strategy are shown in Appendix B.**

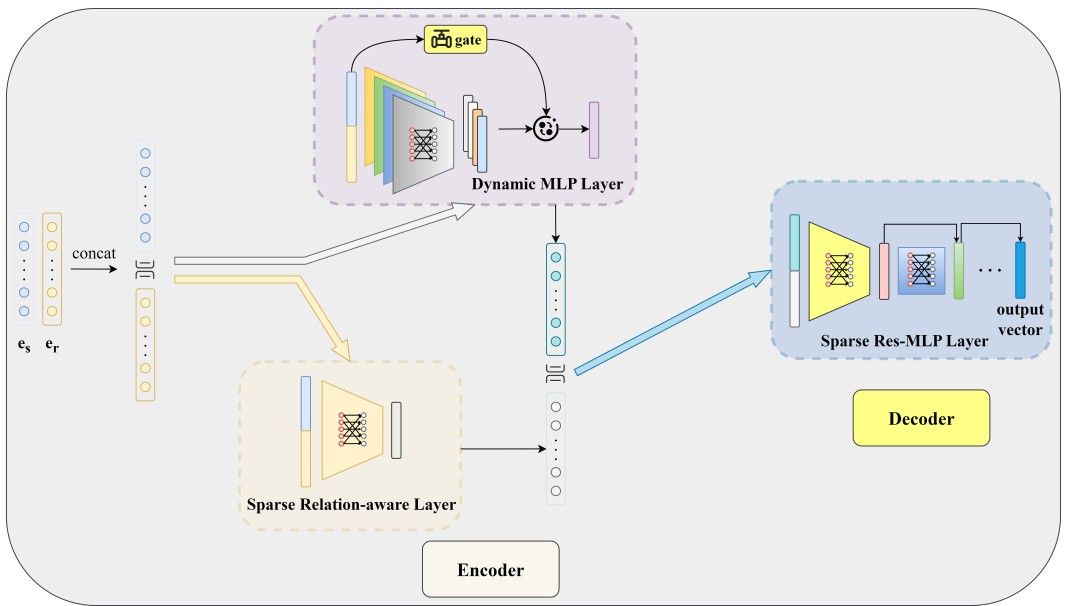

Figure 1: **The architecture of DSparsE.** The model consists of two parts: the encoding end and the decoding end. The encoding end is composed of a dynamic sparse MLP layer and a sparse relation-aware layer. The decoding end is composed of several sparse residual layers.

## 3 METHODOLOGY

This paper proposes a new neural network model that reduces overfitting and improves network robustness through network sparsification and dynamicization. On the one hand, this maximizes the network's expressive power, and on the other hand, it allows the network to deepen and improve its performance ceiling. In ComDensE, deepening the network leads to a decrease in accuracy, and this paper introduces residual connections to alleviate this effect.

Specifically, the proposed DSparsE model consists of two main parts: encoding and decoding. The encoding part includes an MLP layer with $k$ expert blocks (i.e structurally similar MLP sub-blocks (Jacobs et al., 1991)) and a relation-aware MLP layer. The decoding part consists of $m$ levels of residual MLP connection layers. In contrast to conventional dense layers, each module introduces a sparsity degree as a measure of its sparsity. The sparsity degree is a hyperparameter that can be adjusted according to the dataset. The architecture of DSparsE is shown in Figure 1. Specifically, DSparsE takes the $d$-dimensional head node embedding of a knowledge triplet and the $d$-dimensional embedding of the relation as inputs, which are denoted by $e_s$ and $e_r$, respectively. We first concatenate the two embeddings to form a $2d$-dimensional vector, which is passed through a dynamic MLP layer and a sparse relation-aware MLP layer. Each of these modules obtains a $d$-dimensional feature, and the two features are concatenated and passed through an MLP layer to obtain a $d$-dimensional vector $e_{encode}$ as the output of the encoding layer, which can be written as

$$e_{encode} = f(\Omega_P^{\alpha_P}[f(\Omega_R^{\alpha_R}([e_s; e_r])); f(\Omega_D^{\alpha_D}([e_s; e_r]))]) \tag{1}$$

This $d$-dimensional vector is further passed through a decoding layer consisting of residual MLP blocks to obtain the output vector $e_{decode} = \Omega_{Res}^{\alpha_{Res}}(e_{encode})$. After computing the dot product of this vector with the node embedding vector in the embedding table and applying the sigmoid function, the score is obtained as

$$\psi = \sigma(\Omega_{Res}^{\alpha_{Res}}(f(\Omega_P^{\alpha_P}[f(\Omega_R^{\alpha_R}([e_s; e_r])); f(\Omega_D^{\alpha_D}([e_s; e_r]))]))e_o) \tag{2}$$

where $\Omega_D^{\alpha_D}, \Omega_R^{\alpha_R}, \Omega_P^{\alpha_P}, \Omega_{Res}^{\alpha_{Res}}$ denote the dynamic MLP layer, relation-aware MLP layer, projection layer and residual MLP layer, respectively. $\alpha_D, \alpha_R, \alpha_P, \alpha_{Res}$ denote the sparsity degree of the corresponding layer. $\sigma$ denotes the sigmoid function.

## 3.1 DYNAMIC MLP LAYER

The dynamic MLP layer is a linear layer with dynamically changing weights. It can be posited that the network weights change in response to variations in the input (Figure 2 shows the architecture of the dynamic MLP layer). Such structure can enhance the robustness of the model, leading to improved predictive performance. The dynamic layer takes an input vector $\boldsymbol{e}_{in}$ and produces $k$ different output vectors $\boldsymbol{e}_{out_1}, \boldsymbol{e}_{out_2}, ..., \boldsymbol{e}_{out_k}$ by passing the input vector through $k$ different MLP layers. The output vector $\boldsymbol{e}_{out}$ of the dynamic layer is obtained by taking a weighted sum of these output vectors. The combination weights are determined by another small network $g(\cdot)$, which includes a dense fully connected layer and a softmax layer with temperature $T$ as a parameter. The whole process can be formulated as follows:

$$\boldsymbol{e}_{out_i} = \Omega_{D_i}^{\alpha_D}(\boldsymbol{e}_{in}) \tag{3}$$

$$g(\boldsymbol{e}_{in}) = \text{softmax}(\Omega_{gate}(\boldsymbol{e}_{in})/T) \tag{4}$$

$$\boldsymbol{e}_{out} = \Omega_D^{\alpha_D}(\boldsymbol{e}_{in}) = \sum_{i=1}^{k} g(\boldsymbol{e}_{in})_i \cdot \boldsymbol{e}_{out_i} \tag{5}$$

where $\Omega_{gate}, \Omega_{D_i}^{\alpha_D}$ denote the gate network, the $i$-th dynamic MLP layer. $T$ denotes the temperature parameter. $(\cdot)_i$ denotes the $i$-th element of a vector. $\boldsymbol{e}_{out_i}, \boldsymbol{e}_{in}, \boldsymbol{e}_{out}$ denotes the $i$-th output vector, input vector and output vector of the dynamic MLP layer, respectively.

## 3.2 RELATION-AWARE MLP LAYER

When the relation in a knowledge triplet changes, the interaction pattern between the head and tail nodes also changes (Ji et al., 2015; Kim & Baek, 2022). To achieve more accurate feature extraction in the encoding stage, it is necessary to introduce a layer that changes dynamically with the input relation. This can be viewed as part of the network's dynamic nature (See Kim & Baek (2022) for more details). The whole process can be formulated as

$$\boldsymbol{e}_{out} = \Omega_R^{\alpha_R}([\boldsymbol{e}_s; \boldsymbol{e}_{r_i}]) = \boldsymbol{W}_{r_i}^{\alpha_R}[\boldsymbol{e}_s; \boldsymbol{e}_{r_i}] + \boldsymbol{b}_{r_i} \tag{6}$$

where $\boldsymbol{W}_{r_i}^{\alpha_R}, \boldsymbol{b}_{r_i}^{\alpha_R}$ denote the parameter of the $i$-th relation-aware MLP layer. $\boldsymbol{e}_{r_i}$ denotes the embedding of the $i$-th relation. $\boldsymbol{e}_{out}$ denotes the output vector of the relation-aware MLP layer.

## 3.3 RESIDUAL CONNECTION

As shown in Figure 3, a residual block consists of a sparse MLP layer, a batchnorm layer, an activation layer (ReLU or others), a dropout layer and a residual connection. Note that the input and output dimension of the residual block should be the same. The decoder of DSparsE is a stack of $m$ residual blocks with the $i$-th the residual block formulated as

$$\boldsymbol{e}_{Res_i} = f(\text{BN}(\boldsymbol{W}_{Res_i}^{\alpha_{Res}}(\boldsymbol{e}_{Res_{i-1}}) + \boldsymbol{b}_{Res_i}) + \boldsymbol{e}_{Res_{i-1}}) \tag{7}$$

where BN, $\boldsymbol{W}_{Res_i}^{\alpha_{Res}}, \boldsymbol{b}_{Res_i}, \boldsymbol{e}_{Res_i}$ denote the batchnorm layer, the weight and bias of the sparse MLP of the $i$-th residual block, and the output vector of the $i$-th residual block, respectively. Note that $\boldsymbol{e}_{Res0}$ denotes the output of the projection layer.

As introduced in (He et al., 2016; Ma et al., 2022), residual connections help enhance the trainability of deep networks and to some extent prevent deep networks from being outperformed by shallow networks. Similar to the structure of residual convolutional neural networks that can promote network learning, the results in 4.4.3 show that the residual MLP also performs well in link prediction tasks.

## 3.4 SPARSE STRUCTURE

The parameter count of a convolutional block is relatively lower than that of a conventional fully connected layer, because a convolutional layer is essentially a sparse and parameter-sharing linear layer. This result in insufficient information exchange and difficulty in effectively extracting features. Moreover, convolving the feature embeddings of nodes and relations does not have a natural

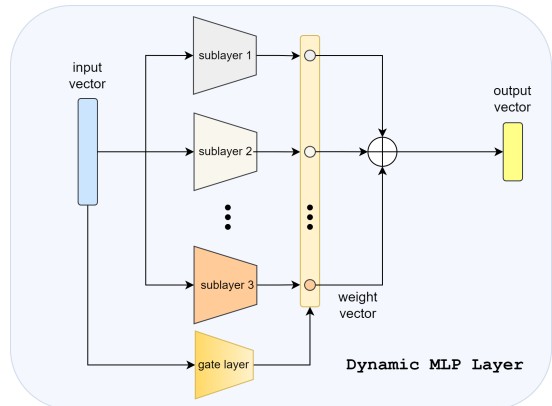
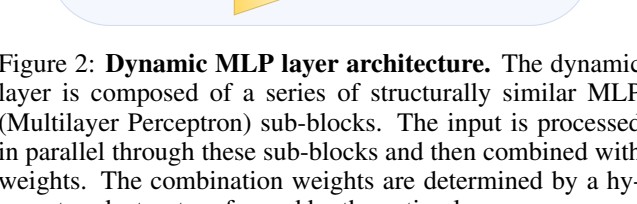
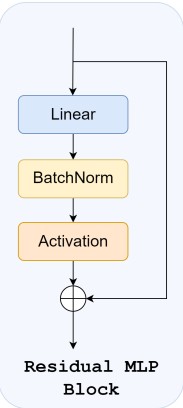

Figure 2: **Dynamic MLP layer architecture.** The dynamic layer is composed of a series of structurally similar MLP (Multilayer Perceptron) sub-blocks. The input is processed in parallel through these sub-blocks and then combined with weights. The combination weights are determined by a hypernetwork structure formed by the gating layer.

Figure 3: **Residual connection structure.** A residual MLP block consists of a sparse MLP layer, a batchnorm layer, an activation layer (ReLU or others), and a residual connection.

physical interpretation. Dense layers, on the other hand, can extract the most features, but they have a large number of useless parameters, poor model generalization, and increased training difficulty (Dettmers & Zettlemoyer, 2019). To tackle this issue, the present study introduces sparsity to the network itself. In the initialization stage of training, the weight matrix is randomly set to zero with a certain probability, forming a sparse MLP layer. A sparse MLP layer can be derived from two directions. Firstly, it can be viewed as the result of enhancing interaction by using a convolutional layer and removing weight sharing. Secondly, it can be viewed as the result of pruning a dense layer (See Appendix C for detail). Given the parameter of a normal dense MLP layer $W, b$ and a sparsity degree $\alpha$, we substitute $W$ with $W^\alpha$, where $W^\alpha$ is formulated as

$$W_{i,j}^\alpha = \begin{cases} 0, & \text{with probability } \alpha \\ W_{i,j}, & \text{with probability } 1 - \alpha \end{cases} \tag{8}$$

while $b$ remains unchanged. Note that $\alpha$ is a hyper parameter that can be adjusted according to the dataset.

## 4 EXPERIMENTS AND ANALYSIS

### 4.1 TRAINING STRATEGY

In our experiments, we use 1-N training strategy introduced by Dettmers et al. (2018) to train DSparsE and adopt the binary cross entropy loss function given by

$$\mathcal{L} = -\frac{1}{N} \sum_i y_i \log \psi(s, r, o_i) + (1 - y_i) \log(1 - \psi(s, r, o_i)) \tag{9}$$

where $N$ is the number of negative samples, $y_i$ is the label of the $i$-th negative sample, and $\psi(s, r, o_i)$ is the score of the $i$-th node as an object. The label $y_i$ of all the entity $o_i$ such that $(s, r, o_i) \in \mathcal{G}$ is 1, and such that $(s, r, o_i) \notin \mathcal{G}$ is 0.

### 4.2 DATASETS AND EVALUATION SETTINGS

We evaluate the performance of DSparsE on two typical datasets: FB15k-237 (Toutanova & Chen, 2015), WN18RR (Dettmers et al., 2018) and YAGO3-10 (Suchanek et al., 2007). Our evaluation of link prediction is conducted in the filtered setting, where we calculate scores for all other potential triples in the test set that are not present in the training, validation, or test set. To generate these potential triples, we corrupt the subjects for object prediction. We use Mean Reciprocal Rank (MRR)

Table 1: **Results on FB15k-237, WN18RR, and YAGO3-10.** The best results are in **bold**, and the second best results are underlined.

| Model | FB15k-237 | | | WN18RR | | | YAGO3-10 | | |
|---|---|---|---|---|---|---|---|---|---|
| | Hits@1 | Hits@10 | MRR | Hits@1 | Hits@10 | MRR | Hits@1 | Hits@10 | MRR |
| TransE (Bordes et al., 2013) | 0.199 | 0.471 | 0.290 | 0.422 | 0.512 | 0.465 | – | – | – |
| TransD (Ji et al., 2015) | 0.148 | 0.461 | 0.253 | – | 0.508 | – | – | – | – |
| DistMult (Yang et al., 2014) | 0.155 | 0.419 | 0.241 | 0.390 | 0.490 | 0.430 | 0.240 | 0.540 | 0.340 |
| CompGCN (Vashishth et al., 2019) | 0.264 | 0.535 | 0.355 | **0.443** | **0.546** | **0.494** | – | – | – |
| R-GCN (Schlichtkrull et al., 2018) | 0.151 | – | 0.249 | – | – | – | – | – | – |
| ConvE (Dettmers et al., 2018) | 0.237 | 0.501 | 0.325 | 0.400 | 0.520 | 0.430 | 0.350 | 0.620 | 0.440 |
| ConvKB (Nguyen et al., 2017) | – | 0.517 | **0.396** | – | 0.525 | 0.248 | – | – | – |
| TuckER (Balažević et al., 2019) | 0.266 | 0.544 | 0.358 | **0.443** | 0.526 | 0.470 | – | – | – |
| ComplEx (Trouillon et al., 2016) | 0.158 | 0.428 | 0.247 | 0.410 | 0.510 | 0.440 | 0.26 | 0.55 | 0.360 |
| RESCAL (Nickel et al., 2011) | 0.269 | 0.548 | 0.364 | 0.417 | 0.487 | 0.441 | – | – | – |
| RotatE (Nickel et al., 2011) | 0.241 | 0.533 | 0.338 | 0.417 | 0.552 | 0.462 | 0.402 | 0.670 | 0.495 |
| KG-BERT (Yao et al., 2019) | – | 0.420 | – | – | 0.524 | – | – | – | – |
| ComDensE (Kim & Baek, 2022) | 0.265 | 0.536 | 0.356 | 0.440 | 0.538 | 0.473 | – | – | – |
| InteractE (Vashishth et al., 2020) | 0.263 | 0.535 | 0.354 | 0.430 | 0.528 | 0.463 | 0.462 | 0.687 | 0.541 |
| **DSparsE** (proposed) | **0.272** | **0.551** | 0.361 | **0.443** | 0.539 | 0.474 | **0.464** | **0.690** | **0.544** |

and Hits at N (Hits@N) metrics to evaluate the performance of our models on these datasets. To ensure robust evaluation, we train and evaluate our models five times and average the performance results. The detail of datasets and experiment settings are given in Appendix F.

## 4.3 OVERALL RESULTS

The overall results are shown in Table 1. It can be seen that DSparsE reached state-of-the-art performance on FB15k-237, WN18RR and YAGO3-10 among neural based models. On FB15k-237, it achieves a improvement of 2.3% and 3.0% on Hits@1 compared to ComDensE and InteractE, respectively. On WN18RR, the improvement is not significant compared to CompGCN and TuckER, but it still outperformed those models based on translation and deep learning. On YAGO3-10, DSparsE achieves a state-of-the-art performance on all the matrices, which highlights the effectiveness of the proposed model. Specifically, DSparsE performs better than those models based on feature convolution. For instance, it achieves a improvement of 14% on Hits@1 compared to ConvE and 6.6% on Hits@10 compared to ConvKB. KG-BERT, which is a link prediction model based on BERT pre-trained language model, performs average on small knowledge graph like FB15k-237 and WN18RR, and its accuracy is much lower than DSparsE. It is observed that DSparsE outperforms KG-BERT, a model based on pretrained language model, with a 24% and 3% improvement on hits@10 on the first two datasets, respectively. For more detailed results, please refer to Appendix H.

## 4.4 ABLATION STUDIES & FURTHER EXPERIMENTS

### 4.4.1 THE EFFECT OF SPARSITY DEGREE

The sparsity of neural networks has a significant impact on their performance. Starting from the fully MLP-based model ComDensE and InteractE, we observe the effect of sparsity on the neural network models and CNN based models. According to Figure 4, the accuracies of both ComDensE and DSparsE models first increase and then decrease with the increase of sparsity, and the highest accuracies of both models appear at a sparsity of 0.5, which refers to the fact that low sparsity in the network can lead to overfitting, limiting its potential, while appropriately increasing sparsity can mitigate this issue. Excessively high sparsity, however, reduces the number of effective parameters and disrupts neuron connections, diminishing

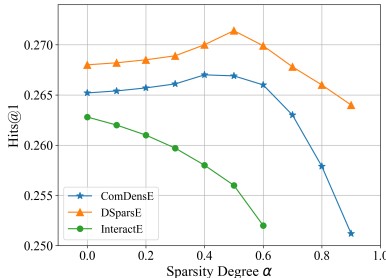

Figure 4: **Hits@1** of InteractE, Com-DensE, and DSparsE on FB15k-237 with different sparsity degree.

the network's expressive power and impairing training due to decreased neuronal interaction. Notably, DSparsE is less adversely affected by increased sparsity compared to ComDensE, owing to its marginally greater parameter count and more complex structure.

On the other hand, the performance of the InteractE model demonstrates a consistent decrease with increasing levels of sparsity. This trend is attributed to the model architecture of InteractE, where only the final feature decoding layer is an MLP layer. Experimental results indicate that introducing increased sparsity over the sparse interactions already captured by the earlier convolutional layers adversely impacts the model's predictive performance.

The results demonstrate that enhancing a network's effectiveness can be achieved by introducing random sparsity. However, this raises two questions:

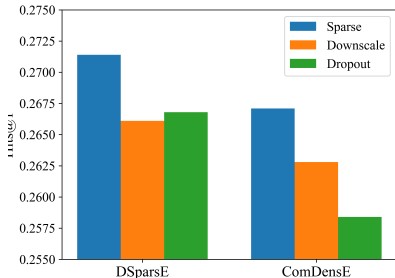

1. **Can actively reducing the scale of the linear layer yield a similar effect?**

2. **Can a similar outcome be achieved by actively increasing the Dropout probability?**

To address these queries, we actively implemented two modifications to the network. Firstly, we actively reduce the output dimension in the linear layers to $\alpha$ times their original number, ensuring the same level of interaction. Specifically, for a linear layer with output dimension $d$, we set the output dimension to $\hat{d} = \alpha d$. Secondly, we actively increase the dropout rate to $\hat{p} = p + \alpha(1 - p)$, where $p$ is the original dropout rate . The results are shown in Figure 5. It indicates that actively decreasing

Figure 5: **The effect after processing the network in different ways.** *Sparse* represents the proposed sparse structure, *Downscale* means cutting off part of the output dimension of the network, and *Dropout* means adding extra dropout based on the original dropout layer.

the number of neurons significantly reduced performance, whereas actively increasing dropout rate drastically deteriorated the final outcomes. This is due to the fact that reducing the neuron number confines the output to a smaller subspace, limiting expressive freedom. Simultaneously, since each training iteration changes the dropout mask, an excessively high dropout actively introduces more uncertainty, thus diminishing network stability.

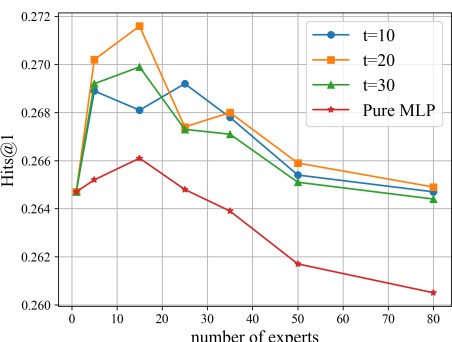

Figure 6: **Hits@1 of DSparsE on FB15k-237 with different number of experts and different temperatures.** $t$ denotes the temperature and *Pure MLP* denotes an MLP layer which has the same number of parameters as the dynamic layer.

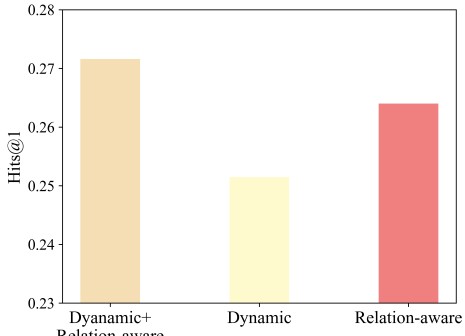

Figure 7: **The effect of Dynamic layer and Relation-aware layer**. The result shows a significant difference when removing the dynamic layer and the relation-aware layer, indicating that both layers are essential to the model. The experiment is conducted on FB15k-237.

### 4.4.2 THE EFFECT OF EXPERTS

Incorporating expert blocks into the dynamic MLP Layer, we observe enhanced dynamics and improved generalization capabilities within the network. Figure 6 illustrates the variation in Hits@1

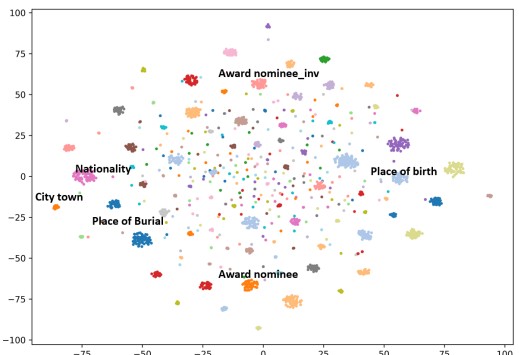 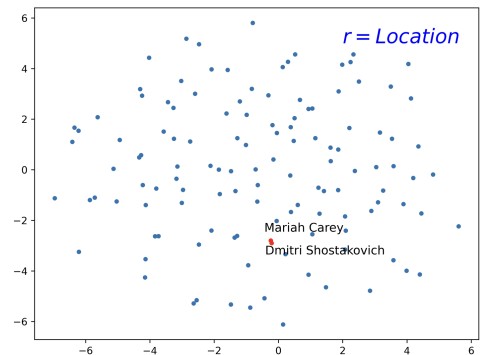

Figure 8: **The output of the gated layer for each entity-relation pair.** Each point represents a unique entity-relation pair in latent space after tSNE reduction. The color of the point represents the relation type.

Figure 9: **The distribution of different entities in the same relation cluster (e.g. a relation named *Location*).** The points that close to each other are somehow semantic smilar in latent space.

scores, contingent upon the number of expert blocks and different temperature settings. The experimental results indicate that the predictive performance initially increases and then decreases with the rising number of expert blocks. Initially, the addition of expert blocks can effectively enhance feature fusion capabilities. This can be explained from two perspectives. Firstly, in contrast to a non-partitioned fully connected structure (i.e., a very wide fully connected layer), the expert blocks in the dynamic layer represent a form of regular sparse connections (See Appendix D for details). These sparse connections are further integrated through a decision layer, namely a gating layer, forming a hypernetwork structure, which brings robustness to the entire network. Secondly, the expert blocks in the dynamic layer can be viewed as sub-modules in an ensemble learning framework. This ensemble learning architecture can effectively suppress the propagation of errors, reducing the variance in prediction results. Under the hypernetwork's constraints, the multi-modular architecture can evolve towards the optimal direction. Experimental results demonstrate that different expert blocks can directly extract features at various levels for node and relation embeddings and predict links through different pathways (See Appendix E for more detail).

However, when the number of expert blocks becomes excessive, the performance deteriorates, which can be attributed to two factors. First, an increase in network parameters introduces additional training complexity, diminishing the network's generalization performance. Second, the gating network is fundamentally a multi-classifier. An excessive number of categories increases the decision-making complexity of the network, making it more prone to difficulties.

Another important influencing factor is the temperature of the dynamic layer. High temperature values make the weight combinations tend towards an average, leading to weight homogenization. Conversely, low temperature values can render many experts ineffective in learning, thus impacting the results.

DSparsE concatenates outputs from the Dynamic and Relation-aware layers, which our further ablation studies show are both essential for optimal performance when used in tandem (Shown in Figure 7). The Dynamic layer compensates for the Relation-aware layer's lack of interconnectedness, facilitating the integration of diverse relational knowledge. The expert layer's gating output is determined by head-relation pairs, fostering a more entity-aware weighting system and enabling the connection of different knowledge types. The interplay between these two layers yields enhanced performance, highlighting their synergistic effect.

Moreover, our investigation into the gating layer's outputs has unveiled some intriguing insights. Each entity-relation pair in the dataset, upon processing through the gating layer, yields an output vector $o$. These high-dimensional vectors were subjected to tSNE reduction, with the resultant visualization displayed in Figure 8 and Figure 9. Each point in this figure represents a unique entity-relation pair, distinguished by varying colors corresponding to different relationships. The visualization result reveals the following observations:

- A tendency for entity-relation pairs of the same relationship type to cluster together, indicating proximity within the output space of the gated layer outputs.

- The spatial distribution of clusters is significantly influenced by the nature of the relationships. For instance, relationships denoting inverse meanings (e.g., nominee_inv and nominee) or semantic opposites (e.g., place of birth vs. place of burial) exhibit a tendency to spatially diverge, exhibiting a unique central symmetry characteristic in the reduced dimensional space, Conversely, relationships with similar semantics (e.g., nationality and city town) are observed to be proximate in the latent space. This proved that DSparsE can capture various associations between entities and relations.

- Alterations in the head entity of a relation pair result in minor shifts within the vector output, confined to a limited scope. Within a fixed relation, the relative positioning of nodes within its corresponding cluster does not display a discernible pattern. This phenomenon can be attributed to the relatively lower frequency of triples involving individual nodes compared to those associated with a particular relation type, posing challenges in accurately modeling semantic information for nodes (Bordes et al., 2013). Still, certain examples, such as Mariah Carley and Dmitri Shostakovich—both notable in the music domain—demonstrate proximity within clusters pertaining to specific relations.

### 4.4.3 THE EFFECT OF RESIDUAL BLOCKS

We observed during training that the need for deeper decoding layers increases with the scale of the dataset (shown in Appendix F). However, simply stacking MLP layers leads to a rapid decline in performance. Employing residual connections effectively maintains the expressive capacity of the network. The residual block helps deepen the network and improve the trainability of the network. Figure 10 shows the results of Hits@1 with different numbers of residual blocks and of Hits@1 with same parameters but without residual blocks. To address the challenge of deepening the network in ComDensE, DSparseE divides the network into two parts: feature fusion and feature decoding. In the feature fusion encoding end, a sparse and dynamic architecture effectively resolves the issue of

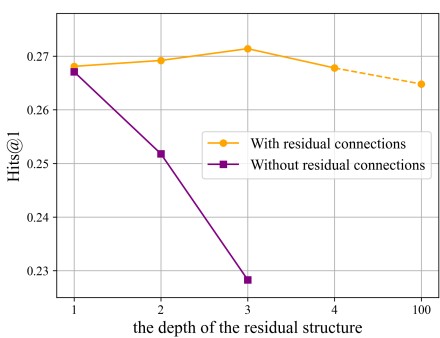

Figure 10: **Hits@1 of DSparsE on FB15k-237 with different number of residual layer depth.** The result deteriorates dramatically with the depth increases if no residual structure are used.

over-tuning in convolutional networks caused by sparse interactions and dense layers. It also enables the network to generate distinct weight responses for different node-relation pairs. To alleviate training pressure and enhance data generalization performance, residual connections are introduced in the MLP layer of the decoding end. The presence of residual connections allows for further deepening of the network without compromising performance, thereby ensuring enhanced expressive capability. If the residual blocks are replaced with normal fully connected layers, the accuracy decreases rapidly as the number of layers increases. In our model, the residual layers effectively reduce the adverse effects of increasing the number of layers on training performance.

## 5 CONCLUSION

This paper proposed the DSparsE for graph completion. This is a new link prediction model structure that uses only MLP layers and employs sparse and residual structures to alleviate overfitting, reduce the difficulty of training deep networks, and improve prediction performance. By introducing expert blocks to construct a dynamic MLP layer, the model's representation power was effectively enhanced. Furthermore, we discovered that the hypernetwork structure formed by gated layers can effectively capture the semantic features and semantic associations of entity-relation pairs, with the results of latent space dimensionality reduction exhibiting interesting clustering and intra-cluster deviation phenomena. The experimental results demonstrate that DSparsE achieves the best performance across multiple metrics on three general datasets FB15k-237, WN18RR, and YAGO3-10. Yet, future work is still required to further explore the patterns of network feature extraction.

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

## A    RELATED WORKS

- **Tensor decomposition models**

  Tensor decomposition models treat the link prediction task as a process of tensor decomposition. It encodes the knowledge graph as a three-dimensional tensor, which is incomplete due to the incompleteness of the knowledge graph. Before learning, the tensor is decomposed into a combination of low-dimensional vectors, from which embeddings of entities and relations can be obtained. The model learns the relationships between vectors by setting a loss function and predicts the existence and correlation of underlying facts in the knowledge graph. Some typical tensor decomposition models include DistMult (Yang et al., 2014), ComplEx (Trouillon et al., 2016), TuckER (Balažević et al., 2019), etc. Although these models are mostly lightweight and easy to train, they are sensitive to sparse data and have limited modeling capabilities.

- **Translational models**

  Translational models are based on the assumption that the relationship between entities can be represented by the translation of an entity vector. The most representative translational model is TransE (Bordes et al., 2013). This model learns the embeddings of entities and relations by

minimizing the energy function, and predicts the existence of underlying facts in the knowledge graph. Translational models are simple and easy to train, but they are not suitable for modeling symmetric relations and complex relations. To address these issues, TransH (Wang et al., 2014), TransR (Lin et al., 2015), and TransD (Ji et al., 2015) models are proposed to enhance the modeling capability by dynamically mapping entities and relations and suppress the homogenization tendency of embedding vectors. Although these methods achieve good results on some datasets such as WN18, FB15K, and FB13, the accuracy is still relatively low. Moreover, the improved methods based on TransE introduce additional computational overhead.

- **Deep learning models**

  Deep learning models for link prediction can be roughly categorized as :

  1. Models based on simple fully connected layers (MLPs)
  2. Models based on convolutional neural networks (CNNs)
  3. Models based on graph neural networks (GNNs)
  4. Models based on recurrent neural networks (RNNs) or transformers.

  During link prediction, these networks typically take known node and relation embeddings as input (some networks also take additional information as input), and obtain a result vector after encoding and decoding the input data through several neural network linear and nonlinear layers. Networks based on simple fully connected layers, such as ComDensE (Kim & Baek, 2022), add a Relation-aware component, which generates different weight matrices for different relations that appear in the training set based on the common layer. Models based on convolutional neural networks (CNNs) include ConvE (Dettmers et al., 2018), ConvKB (Nguyen et al., 2017), ConvR(Jiang et al., 2019), and InteractE (Vashishth et al., 2020). These methods convert embedding vectors into two-dimensional feature maps in different ways and apply filters for convolution. More specifically, ConvR uses the relation embedding vector as the convolution kernel, while InteractE enhances the interaction between features by reshaping them into a checkerboard pattern. Models based on graph neural networks (GNNs) include R-GCN (Schlichtkrull et al., 2018) and CompGCN (Vashishth et al., 2019). These methods use graph convolutional networks to grab the neighborhood information of entities and relations and aggregate them into the entity embedding vector. Those method natually take advantage of the graph structure and achieve good results on some datasets. However, the parallelization challenge caused by the heterogeneous graph structure limits the performance of these methods. Some methods achieve link prediction by fine-tuning pre-trained language models, such as KG-BERT (Yao et al., 2019) and Rhelphormer (Bi et al., 2022). Although these models achieve good results, they suffer from high complexity and require external information beyond the knowledge graph (encyclopedia entries for instance).

## B  Scoring functions

Most link predition models based on machine learning methods consists of two components: an embedding layer and a scoring function. The embedding layer maps the entities $s, o$ and relations $r$ in the knowledge graph to a low-dimensional vector space $\mathbb{R}^d$, and the scoring function calculates the score of the triple based on the embeddings of the entities and relations. The widely used scoring functions include several kind of distance-based functions, such as the $l_1$ distance, $l_2$ distance, and cosine distance, and the bilinear function. The distance-based functions are usually used in translational models, while the bilinear function is usually used in tensor decomposition models. Some typical scoring functions are shown in Table 2.

## C  Intuitive explanation of the sparse structure

Figure 11 shows the intuitive process of the sparse structure derived from both convolutional and dense layers.

## D  Why dynamic layer is actually a sparse layer

Figure 12 shows the comparison between a fully connected layer and a connected layer with expert blocks. Let $x$ be the input vector, $W$ and $W_i$ be the weight matrix and the weight matrix of the $i$-th

Table 2: **Scoring functions.** The notations $e_s, e_r, e_o$ are the embeddings of the subject, relation, and object, respectively. $w_r$ is the normal vector of the hyperplane of the relation $r$. $\omega$ is the convolution kernel. $W$ is the weight matrix of the linear layer. $f$ denotes the activation function. $\mathcal{P}_k$ denotes the $k$-th permutation of the concatination of the input feature vector. $\phi$ denotes the reshaping operator. $\Omega$ is the common projection function. $\Omega_r$ is the relation-specific projection function. $\circledast$ is the circular convolution operator. vec is the vectorization operator. $[\cdot; \cdot]$ is the concatenation operator. $[\cdot]_{1d}$ is the reshaping operator. The notation $\langle e_s, e_r, e_o \rangle = e_s^T(e_r \circ e_o)$, where $\circ$ is the Hadamard product operator. $\Re(\cdot)$ is the real part operator.

| Model | Scoring functions $\psi(e_s, e_r, e_o)$ |
| --- | --- |
| TransE (Bordes et al., 2013) | $\|e_s + e_r - e_o\|_p$ |
| TransH (Wang et al., 2014) | $\left\|(e_s - w_r^\top e_s w_r) + e_r - (e_o - w_r^\top e_o w_r)\right\|_p$ |
| DistMult (Yang et al., 2014) | $\langle e_s, e_r, e_o \rangle$ |
| ComplEx (Trouillon et al., 2016) | $\Re(\langle e_s, e_r, e_o \rangle)$ |
| ConvE (Dettmers et al., 2018) | $f\left(\text{vec}\left(f\left([\overline{e_s}; \overline{e_r}] * \omega\right)\right) W\right) e_o$ |
| InteractE (Vashishth et al., 2020) | $f\left(\text{vec}\left(f\left(\phi\left(\mathcal{P}_k\right) \circledast \omega\right)\right) W\right) e_o$ |
| ComdensE (Kim & Baek, 2022) | $f\left(\left[\left[f\left(\Omega_r\left([e_s; e_r]_{1d}\right)\right); f\left(\Omega\left([e_s; e_r]_{1d}\right)\right)\right]_{1d}^\top W\right) e_o$ |

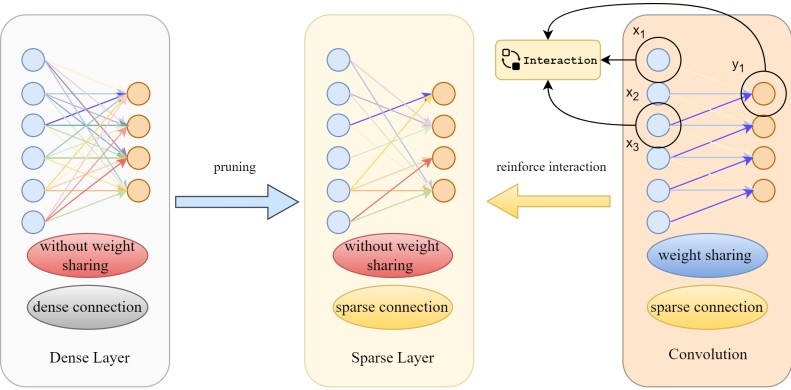

Figure 11: **The intuitive explanation of sparse structure.** The sparse layer can be regarded either as a fully connected layer with unstructed pruning or as a convolution layer with enhanced feature interaction with no weight sharing.

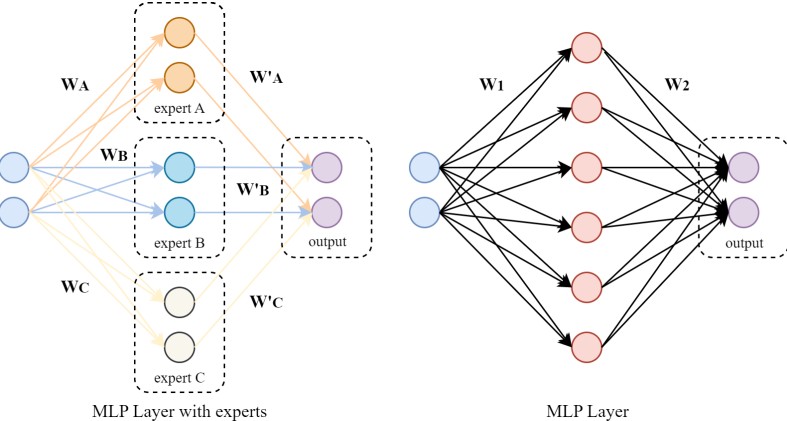

Figure 12: **Comparison between a fully connected layer and a connected layer with expert blocks.** The Dynamic layer can be conceptualized as a structured sparsity in the neural connections between the hidden layers and the output layer. **Note that the activation process are omitted for clarity.**

expert block, respectively. Let $b$ be the bias vector, and $y$ be the output vector. The output of the fully connected layer with one hidden layer can then be given by the following equation:

$$y = W_2 f(W_1 x + b_1) + b_2 \tag{10}$$

While the output of the connected layer with $n$ expert blocks can be given by the following equation:

$$y = \sum_{i=1}^{n} \omega_i f(W_i x + b_i) \tag{11}$$

which can be reformulated as:

$$y = W' \operatorname{concat}(f(W_1 x + b_1); \cdots ; f(W_n x + b_n)) + 0 \tag{12}$$

where the $W'$ is the equivalent weight matrix of the connected layer with $n$ expert blocks, and concat is the operator that concatinate the output of each expert block into a matrix. It can be observed that $W'$ is a well structured sparse matrix. Specifically, the number of non-zero elements in each row is equivalent to the number of expert blocks. When the elements of each row are grouped (with the number of groups equal to the number of expert blocks), each group contains only one non-zero element. Moreover, the position of this non-zero element varies across rows, but it is guaranteed to appear exactly once in each row.

## E    RESPONSE OF THE EXPERT BLOCK TO THE INPUT

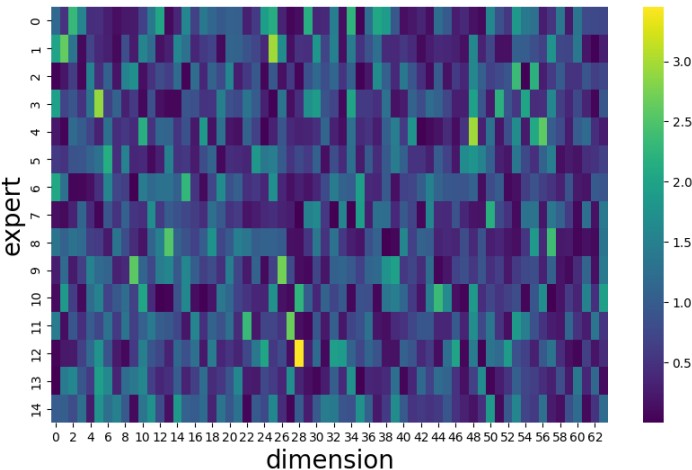

Figure 13: **The response of the expert block to the input.** In the heatmap, the horizontally arranged colored blocks represent the importance of the input neurons. Specifically, a higher brightness indicates a higher normalized weight value corresponding to the input neuron. Note that for ease of viewing, the image shows the inputs after the embeddings of the head and tail nodes have been reduced to only 32 dimensions each.

The importance of a neuron can be represented by its normalized weight, which we define as follows.

**Definition 1** *The normalized weight of a neuron is defined as the absolute value obtained after normalizing the Euclidean norm (L2 norm) of the vector consisting of all weight coefficients associated with it. Let $\omega_n$ denotes the normalized weight of neuron of neuron n,* norm *denotes the norm operator, and $\boldsymbol{\omega_n}$ denotes the vecotor formed by all the weights coefficient to a certain neuron n. This can be specifically given by the following equation:*

$$\omega_n = |\operatorname{norm}(\|\boldsymbol{\omega_n}\|_2)| \tag{13}$$

Figure 13 indicates that different expert blocks have maximal responses to different input neurons, indicating that each input neuron has varying levels of importance to each expert block. Furthermore, each expert block responds maximally to only a few neurons. This suggests that expert blocks are capable of capturing distinct parts of the feature information, thereby reducing the decision errors that might arise from a single layer.

## F  DETAILED EXPERIMENT SETTINGS

During training, we use Adam as the optimizer and search the hyperparameters with learning rate $\in \{0.0001, 0.001, 0.002, 0.005, 0.01\}$, batch size $\in \{128, 256\}$, embedding dimension $\in \{256, 512\}$, sparsity degree $\in \{0.3, 0.4, 0.5, 0.6, 0.7, 0.8, 0.9, 0.95\}$, number of experts $\in \{10, 12, 15, 20, 25, 30, 35, 40, 45, 50, 60\}$, number of residual blocks $\in \{1, 2, 3, 4, 5, 10, 50\}$, temperature $\in \{10, 15, 20, 25, 30\}$. We use the best hyperparameters found during the search to train the model for 800 epochs. The best hyperparameters are selected based on the performance on the validation set. The embedding dimension is set to 256 and the sparsity degree is set to 0.5 for all relevent layers. The number of experts is set to 15, 35, 12 for FB15k-237, WNRR18, and YAGO3-10, respectively. The number of residual blocks is set to 3, 1, 5 for FB15k-237, WN18RR, and YAGO3-10, respectively. The temperature is set to 20, the learning rate is set to 0.002, and the batch size is set to 128, respectively.

Table 3: The details of the datasets.

|  | WN18RR | FB15k-237 | YAGO3-10 |
|---|---|---|---|
| #Entity | 40,943 | 14,541 | 123,182 |
| #Relation | 11 | 237 | 37 |
| #Train | 86,835 | 272,115 | 1,079,040 |
| #Valid | 3,034 | 17,535 | 5,000 |
| #Test | 3,134 | 20,466 | 5,000 |

## G  PARAMETER EFFICIENCY

DSparsE incorporates the sparse structure into the fully connected layer and the convolutional layer. Table 5 shows the number of parameters of ComDensE, InteractE and DSparsE. It can be observed that DSparsE has fewer parameters than InteractE and ComDensE on WN18RR18 and slightly more parameters than ComDensE on FB15k-237. This is because the number of experts in DSparsE is larger than the number of relation-specific weight matrices and shared dense layer in ComDensE. However,our experiments conducted on an RTX 4080 laptop indicate that there is no significant difference in the training speed among the three models.

It is important to note that although DSparse employs an unstructured sparse topology that typically does not reduce computational overhead under normal circumstances, for well-designed hardware, this sparse structure can effectively conserve computational resources and enhance processing speed.

Table 4: **The number of parameters of ComDensE, InteractE and DSparsE.**

| Number of parameters | FB15k-237 | WN18RR |
|---|---|---|
| InteractE | 18M | 60M |
| ComDensE | 66M | 33M |
| DSparsE | 69M | 29M |

Table 5: **The average running time per epoch of ComDensE, InteractE and DSparsE on FB15k-237.** The device used for the experiment is NVIDIA GeForce RTX 4080 Laptop GPU.

| model | DSparsE | ComDensE | InteractE |
|---|---|---|---|
| **FB15k237** | 2.67s | 2.61s | 2.58s |
| **WN18RR** | 3.44s | 3.60s | 3.43s |

## H  DETAILED RESULTS

Table 6 shows the detailed results for each relation category.

Table 6: **Detailed results** of 4 different kinds of relations 1:1, 1:N, N:1, N:N on FB15k-237. Note that a given relation is 1:1 if a head can appear with at most one tail, 1:N if a head can appear with many tails, N:1 if many heads can appear with the same tail, or N:N if multiple heads can appear with multiple tails (Bordes et al., 2013).

| Relation Type | | InteractE | | | ComDensE | | | DSparsE | | |
|---|---|---|---|---|---|---|---|---|---|---|
| | | MRR | Hits@10 | Hits@1 | MRR | Hits@10 | Hits@1 | MRR | Hits@10 | Hits@1 |
| Pred Head | 1:1 | 0.386 | 0.547 | 0.245 | 0.422 | 0.557 | 0.349 | **0.434** | **0.572** | **0.358** |
| | 1:N | **0.106** | **0.192** | 0.043 | 0.084 | 0.181 | 0.043 | 0.101 | 0.185 | **0.044** |
| | N:1 | 0.466 | 0.647 | 0.369 | 0.466 | 0.649 | 0.372 | **0.467** | **0.655** | **0.376** |
| | N:N | 0.276 | 0.476 | 0.164 | 0.279 | 0.476 | 0.187 | **0.287** | **0.494** | **0.195** |
| Pred Tail | 1:1 | 0.368 | 0.547 | 0.229 | 0.422 | 0.563 | 0.349 | **0.428** | **0.570** | **0.351** |
| | 1:N | 0.777 | 0.708 | **0.881** | **0.779** | 0.884 | 0.717 | 0.778 | **0.886** | 0.796 |
| | N:1 | 0.074 | 0.141 | 0.034 | 0.084 | 0.169 | **0.043** | **0.088** | **0.171** | 0.042 |
| | N:N | 0.395 | 0.617 | 0.272 | **0.396** | 0.618 | 0.285 | 0.395 | **0.624** | **0.286** |

