# OpenReview forum: "DSparsE: Dynamic Sparse Embedding for Knowledge Graph Completion"
_ICLR.cc/2024/Conference — Submitted to ICLR 2024_

### Official Review · Reviewer_uERB · 2023-10-30

**Soundness:** 2 fair
**Presentation:** 2 fair
**Contribution:** 2 fair
**Rating:** 3
**Confidence:** 4

**Summary:**

This paper proposes a knowledge graph link prediction model DSparsE, in which the dynamic layer and residual structure are incorporated to achieve higher efficiency. Experimental results on two datasets show its effectiveness.

**Strengths:**

1. The motivation of this paper is easy to understand.
2. The proposed model seems technically reasonable.

**Weaknesses:**

1. The difference and the superiority of DSparsE compared with baseline models are unclear.
2. The contribution of this paper is limited. It only combines some existing techniques.
3. The current version of this paper is not easy to follow. There are some uncommon words without explanation, such as "reminiscent".
4. The citation and analysis of previous models are insufficient.
5. More datasets are required to verify the effectiveness of the proposed model.
6. More experimental results about the scalability of the proposed model should be provided.

**Questions:**

None

---

> ### Author Response · Authors · 2023-11-19
>
> **We appreciate your valuable comments. We have revised our manuscript and clarified the contributions in the revised version. We sincerely hope you will find the time to review our improved manuscript and look forward to your valuable comments.**
>
>
> ---
>
> >* **W1: The difference and the superiority of DSparsE compared with baseline models are unclear.**
> >* **W2: The contribution of this paper is limited. It only combines some existing techniques.**
>
> **Response**
> Thank you for your comment. The key advantages of DSparsE can be summarized in three points:
>
> 1. Dynamic Characteristics: The relation-aware layer in ComDensE[1] focuses only on the dynamic nature of network weights as relationships change, neglecting the dynamics associated with node changes. Indeed, designing a separate weight matrix for each node would lead to an explosive increase in parameters with the scale of the graph. Therefore, we adopt a hypernetwork structure to build a network where parameter weights can change with the input entity-relation pairs. This network can consider both entity and relation node information, and works well in conjunction with the relation-aware layer. Additional ablation studies in the revised version of the paper prove that parallel dynamic and relation-aware layers can achieve better performance.
>
>     | Dynamic+Relation-aware | Dynamic layer only | Relation-aware layer only |
>     |:-------:|:-------:|:-------:|
>     | **0.272**| 0.252 | 0.264 |
>
> 2. Sparse Characteristics: Traditional neural networks face issues of overfitting and generalization, which are more severe in knowledge graph link prediction models based on neural networks. This is due to the open-world assumption[2] in knowledge graphs, meaning the graphs are incomplete and the dataset knowledge is extremely limited compared to the actual distribution of knowledge. Thus, our model introduces a fixed sparse structure, different from the dropout feature, which is determined at model initialization. Our inspiration comes from convolutional neural networks like ConvE[3] and InteractE[4], which have sparse connections with weight sharing. We consider random sparse connections between neurons without weight sharing and use a hyperparameter alpha to control the density of feature interactions. Further experiments also show that sparse connections on top of dropout can optimize the network model's predictive performance. The results below show the effectiveness of sparse structure. Note that extra dropout represents setting dropout rate $\hat{p}$ $=$ $p + \alpha(1-p)$, where $\alpha$ denotes the sparsity degree and $p$ denotes the original dropout rate. Here we set $\alpha$ to 0.5. The results indicate that sparse structure can effectivly enhance the performance.
>
> | model | Dropout + sparse | Extra dropout |
> | :---: | :---: | :---: |
> | DSparsE | **0.272** | 0.266 |
> | ComDensE | 0.267 | 0.258 |
>
>
> 3. Residual Characteristics: Existing research has shown that deepening network layers can enhance expressiveness[5], but this comes with convergence difficulties, a problem also present in knowledge graph link prediction models. For example, the authors of ComDensE found that increasing the number of layers actually reduced network performance. We also found that simply adding front-end fully connected layers in matrix decomposition or similar models reduces their expressiveness. However, the only way for shallow models to maintain effectiveness with increasing graph scale is to increase the embedding dimension, leading to greater computational costs. If the embedding dimension is kept constant, performance declines with increasing graph scale (especially the number of entities), such as RESCAL's performance on WNRR18 being worse than on FB15k-237 with the same embedding dimension. In DSparsE, we applied expert layers to maximize feature extraction and fusion, requiring a powerful decoding module. In experiments, we found that 3-5 layers of residual neural networks provided good results on Fb15k-237 and YAGO3-10. To some extent, the introduction of residual layers allows the network to deepen further, maximizing its expressive potential. This is beneficial for large knowledge graphs like YAGO3-10.
> We have provided a table of the optimal number of residual layers for three models, which shows that the depth of the decoding layer needs to increase with the scale of the dataset. The introduction of residual connections helps to stabilize the network training process and maintain the predictive results (the experimental results have been presented in the paper).
>
> | Dataset | Entity | Relation | Train | optimal residual layer depth |
> | :---: | :---: | :---: |:---:|:---:|
> | FB15k-237 | 14,541 | 237 | 272,115 | 3 |
> | WNRR18 | 40,943 | 11 | 86,835 | 1 |
> | YAGO3-10 | 123,182 | 37 | 1079,040 | 5 |

---

> > ### Author Response · Authors · 2023-11-19
> >
> > (Following the previous text) Regarding your second question, indeed, our model consists of some existing modules, and our work is merely a slight improvement upon this foundation. However, it's important to clarify that these modules are organically combined, rather than a haphazard mix.
> >
> > Firstly, the relation-aware layer is only dynamically responsive to relations. We wanted the network to also dynamically respond to entities, motivating us to add a dynamic layer. The introduction of expert blocks significantly increased the density of feature interactions. Consequently, we incorporated a fixed sparse structure to mitigate excessive interaction between features, alleviating overfitting and enhancing model robustness. The decoding side needs to decode massive feature interactions, and we also noted that the need for depth at the decoding end increases with the scale of the dataset. However, for typical linear layers, we observed a rapid deterioration in performance with increasing depth. Naturally, this led us to utilize the characteristics of residual structures to maximize the network's expressive capabilities.
> >
> > Furthermore, by exploring the distribution characteristics of the input entity-relation pairs in the output latent space of the gating hypernetwork layer, we made some interesting observational conclusions. For this point, we invite you to refer to the visual figures in our paper for a better appraisal of our work.
> >
> >
> > ---
> >
> > >* **W3: The current version of this paper is not easy to follow. There are some uncommon words without explanation, such as "reminiscent".**
> > >* **W4: The citation and analysis of previous models are insufficient.**
> >
> > **Response**
> >
> > Thank you for your comment. We have revised these words in our manuscript. Following your suggestions, we have thoroughly revised key sections of the paper, including the abstract and experimental analysis.
> >
> > The sentence you referred to in the abstract has been revised to:
> > >By applying sparse structure to the dynamic MLP layers composed of multiple expert blocks and the residual MLP layers in decoders, DSparsE maintains the network’s robustness while ensuring the capability for feature interaction.
> >
> > We have also made corresponding revisions regarding the citation and analysis of related work. We have added an introduction and relevant references to GCNs (Graph Convolutional Networks)[6, 7], while introducing the Dynamic layer, we have cited the original MoE (Mixture of Experts) paper _Adaptive mixtures of local experts_[8], as well as subsequent works such as _Outrageously large neural networks: The sparsely-gated mixture-of-experts layer_[9]. In the discussion of residual connections, we referred to the seminal paper _Deep residual learning for image recognition_[10]. Should you have any further suggestions, please do not hesitate to enlighten us.
> >
> > ---

---

> > > ### Author Response · Authors · 2023-11-19
> > >
> > > >* **W5: More datasets are required to verify the effectiveness of the proposed model.**
> > > >* **W6: More experimental results about the scalability of the proposed model should be provided.**
> > >
> > > **Response**
> > > Thank you for your constructive suggestions. In the revised version of the paper, we have included a dataset, YAGO3-10. Now we present the experimental results of DSparsE on the YAGO3-10 dataset across various metrics:
> > >
> > > | Dataset | Entity | Relation | Train | Valid | Test |
> > > | :---: | :---: | :---: |:---:|:---:|:---:|
> > > | YAGO3-10 | 123,182 | 37 | 1079,040 | 5000 | 5000 |
> > >
> > >
> > > | Dataset | Hits@1 | Hits@10 | MRR |
> > > | :---: | :---: | :---: |:---:|
> > > | DistMult | 0.240 | 0.540 | 0.340 |
> > > | ConvE | 0.350 | 0.620 | 0.440 |
> > > | ComplEx | 0.260 | 0.550 | 0.360 |
> > > | RotatE | 0.402 | 0.670 | 0.495 |
> > > | InteractE | 0.462 | 0.687 | 0.541 |
> > > | **DSparsE** | **0.464** | **0.690** | **0.544** |
> > >
> > > We also give the detailed results as well as the comparison on FB15k-237:
> > >
> > > |                    |      | MRR   | Hits@10 | Hits@1 | MRR   | Hits@10 | Hits@1 | MRR   | Hits@10 | Hits@1 |
> > > |--------------------|------|-------|---------|--------|-------|---------|--------|-------|---------|--------|
> > > |                    | InteractE  |  |    | ComDensE  |  |    |   | DSparsE |   | |
> > > | **Pred Head**      | 1:1  | 0.386 | 0.547   | 0.245  | 0.422 | 0.557   | 0.349  | **0.434** | **0.572**   | **0.358**|
> > > |                    | 1:N  | **0.106** | **0.192**   | 0.043  | 0.084 | 0.181   | 0.043  | 0.101 | 0.185   | **0.044**|
> > > |                    | N:1  | 0.466 | 0.647   | 0.369  | 0.466 | 0.649   | 0.372  | **0.467** | **0.655**   | **0.376**|
> > > |                    | N:N  | 0.276 | 0.476   | 0.164  | 0.279 | 0.476   | 0.187  | **0.287** | **0.494**   | **0.195**|
> > > | **Pred Tail**      | 1:1  | 0.368 | 0.547   | 0.229  | 0.422 | 0.563   | 0.349  | **0.428** | **0.570**   | **0.351**|
> > > |                    | 1:N  | 0.777 | 0.708   | **0.881** | 0.779 | 0.884   | 0.717  | 0.778 | **0.886**   | 0.796|
> > > |                    | N:1  | 0.074 | 0.141   | 0.034  | 0.084 | 0.169   | **0.043** | **0.088** | **0.171**   | 0.042|
> > > |                    | N:N  | 0.395 | 0.617   | 0.272  | **0.396** | 0.618   | 0.285  | 0.395 | **0.624**   | **0.286**|
> > >
> > > For the original Figure 4, we added a comparison with InteractE, demonstrating the monotonic decline of InteractE with increasing sparsity. We have explained this in the revised version of the paper as follows:
> > >
> > > >On the other hand, the performance of the InteractE model demonstrates a consistent decrease with increasing levels of sparsity. This trend is attributed to the model architecture of InteractE, where only the final feature decoding layer is an MLP layer. Experimental results indicate that introducing increased sparsity over the sparse interactions already captured by the earlier convolutional layers adversely impacts the model's predictive performance.
> > >
> > > While investigating the impact of sparsity, we also conducted partial tests of DSparsE on the WN18RR dataset. The experiments are still ongoing, but we provide some of the data as follows, which is consistent with the analytical conclusions in the original text:
> > >
> > > | Sparsity | Hits@1 |
> > > | :---: | :---: |
> > > | 0. | 0.438 |
> > > | 0.3 | 0.441 |
> > > | 0.5 | 0.443 |
> > > | 0.8 | 0.421 |
> > >
> > > Additional ablation studies in the revised version of the paper prove that parallel dynamic and relation-aware layers can achieve better performance.
> > >
> > > | Dynamic+Relation-aware | Dynamic layer only | Relation-aware layer only |
> > > |:-------:|:-------:|:-------:|
> > > | **0.272**| 0.252 | 0.264 |

---

> > > > ### Author Response · Authors · 2023-11-19
> > > >
> > > > (Following the previous text) The Dropout method applies different masks with each training iteration, which adds to the instability of the training. In our method, the pattern of sparse connections is already determined at initialization. Essentially, our method cuts the connections between neurons, weakening the intensity of feature interactions. Therefore, through empirical evidence, we demonstrate that:
> > > >
> > > > 1. Our method has advantages over simply reducing the network size (i.e., the number of output neurons).
> > > > 2. Our method has advantages over the use of Dropout alone.
> > > >
> > > > So we explore the effects of replacing the sparse structure with additional dropout or reducing the number of neurons, as shown below:
> > > >
> > > > | model | Dropout + sparse | Extra dropout | Downscale |
> > > > | :---: | :---: | :---: |:---:|
> > > > | **DSparsE** | **0.272** | 0.266 | 0.265 |
> > > > | ComDensE | 0.265 | 0.258 | 0.263 |
> > > >
> > > > >The results demonstrate that enhancing a network's effectiveness can be achieved by introducing random sparsity. However, this raises two questions:
> > > >     > 1. __Can actively reducing the scale of the linear layer yield a similar effect?__
> > > >     > 2. __Can a similar outcome be achieved by actively increasing the Dropout probability?__
> > > > >
> > > > >To demonstrate the effectiveness of sparse structure, we actively implemented two modifications to the network. Firstly, we actively reduce the output dimension in the linear layers to $\alpha$ times their original number, ensuring the same level of interaction. Specifically, for a linear layer with output dimension $d$, we set the output dimension to $\hat{d} = \alpha d$. Secondly, we actively increase the dropout rate to $\hat{p} = p + \alpha(1 - p)$, where $p$ is the original dropout rate. The result are shown above. It indicates that actively decreasing the number of neurons significantly reduced performance, whereas actively increasing dropout rate drastically deteriorated the final outcomes. This is due to the fact that reducing the neuron number confines the output to a smaller subspace, limiting expressive freedom. Simultaneously, since each training iteration changes the dropout mask, an excessively high dropout actively introduces more uncertainty, thus diminishing network stability.
> > > >
> > > > Furthermore, we have studied the characteristics of the data distribution in the latent subspace output by the gating layer and have provided interesting visual results：
> > > >
> > > > >Moreover, our investigation into the gating layer's outputs has unveiled some intriguing insights. Each entity-relation pair in the dataset, upon processing through the gating layer, yields an output vector $\bm{o}$. These high-dimensional vectors were subjected to tSNE reduction, with the resultant visualization displayed in Figure 8 and Figure 9. Each point in this figure represents a unique entity-relation pair, distinguished by varying colors corresponding to different relationships. The visualization result reveals the following observations:
> > > > >1. A tendency for entity-relation pairs of the same relationship type to cluster together, indicating proximity within the output space of the gated layer outputs.
> > > > >2. The spatial distribution of clusters is significantly influenced by the nature of the relationships. For instance, relationships denoting inverse meanings (e.g. nominee\_inv and nominee) or semantic opposites (e.g. place of birth vs. place of burial) exhibit a tendency to spatially diverge, exhibiting a unique central symmetry characteristic in the reduced dimensional space, Conversely, relationships with similar semantics (e.g. nationality and city town) are observed to be proximate in the latent space. This proved that DSparsE can capture various associations between entities and relations.
> > > > >3. Alterations in the head entity of a relation pair result in minor shifts within the vector output, confined to a limited scope. Within a fixed relation, the relative positioning of nodes within its corresponding cluster does not display a discernible pattern. This phenomenon can be attributed to the relatively lower frequency of triples involving individual nodes compared to those associated with a particular relation type, posing challenges in accurately modeling semantic information for nodes. Still, certain examples, such as Mariah Carley and Dmitri Shostakovich—both notable in the music domain—demonstrate proximity within clusters pertaining to specific relations.
> > > >
> > > > For the images mentioned in the text, please refer to our revised version of the paper.
> > > >
> > > >
> > > > In addition, we have included some data and charts in the appendix of the revised paper, which will help readers better understand the principles and effects of our model.
> > > >
> > > > ---

---

> > > > > ### Author Response · Authors · 2023-11-19
> > > > >
> > > > > **Thank you for patiently reading our response. Once again, we invite you to review our revised version and look forward to your further comments!**
> > > > >
> > > > >
> > > > > ---
> > > > >
> > > > >
> > > > > [1] Drummond N, Shearer R. The open world assumption[C]//eSI Workshop: The Closed World of Databases meets the Open World of the Semantic Web. 2006, 15: 1.
> > > > >
> > > > > [2] Kim M, Baek S. Comdense: combined dense embedding of relation-aware and common features for knowledge graph completion[C]//2022 26th International Conference on Pattern Recognition (ICPR). IEEE, 2022: 1989-1995.
> > > > >
> > > > > [3] Dettmers T, Minervini P, Stenetorp P, et al. Convolutional 2d knowledge graph embeddings[C]//Proceedings of the AAAI conference on artificial intelligence. 2018, 32(1).
> > > > >
> > > > > [4] Vashishth S, Sanyal S, Nitin V, et al. Interacte: Improving convolution-based knowledge graph embeddings by increasing feature interactions[C]//Proceedings of the AAAI conference on artificial intelligence. 2020, 34(03): 3009-3016.
> > > > >
> > > > > [5] Chatterjee S, Zielinski P. On the generalization mystery in deep learning[J]. arXiv preprint arXiv:2203.10036, 2022.
> > > > >
> > > > > [6] Vashishth S, Sanyal S, Nitin V, et al. Composition-based multi-relational graph convolutional networks[J]. arXiv preprint arXiv:1911.03082, 2019.
> > > > >
> > > > > [7] Schlichtkrull M, Kipf T N, Bloem P, et al. Modeling relational data with graph convolutional networks[C]//The Semantic Web: 15th International Conference, ESWC 2018, Heraklion, Crete, Greece, June 3–7, 2018, Proceedings 15. Springer International Publishing, 2018: 593-607.
> > > > >
> > > > > [8] Jacobs R A, Jordan M I, Nowlan S J, et al. Adaptive mixtures of local experts[J]. Neural computation, 1991, 3(1): 79-87.
> > > > >
> > > > > [9] Shazeer N, Mirhoseini A, Maziarz K, et al. Outrageously large neural networks: The sparsely-gated mixture-of-experts layer[J]. arXiv preprint arXiv:1701.06538, 2017.
> > > > >
> > > > > [10] He K, Zhang X, Ren S, et al. Deep residual learning for image recognition[C]//Proceedings of the IEEE conference on computer vision and pattern recognition. 2016: 770-778.

---

> ### Author Response · Authors · 2023-11-21
> **We are eager for your reply!!**
>
> **We have carefully considered your feedback and updated our manuscript. Could you please take a moment to review our changes and let us know if they resolve your queries? Your feedback is invaluable to us!**

---

> > ### Comment · Reviewer_uERB · 2023-11-23
> > **Thank the authors for response**
> >
> > Thank you for clarifying w2 and providing more experimental results. However, the technique contributions of this paper are still limited, and the superiority of the proposed method in efficiency is still not convincing. Based on my initial review, your response, and the comments of other reviewers. I am inclined to keep my score.

---

### Official Review · Reviewer_S6XC · 2023-10-30

**Soundness:** 2 fair
**Presentation:** 2 fair
**Contribution:** 2 fair
**Rating:** 5
**Confidence:** 3

**Summary:**

This paper proposes DSparsE, a novel knowledge graph link prediction model.  This model introduces a dynamic layer into the encoding end and a residual structure into the decoding end. Moreover, this model achieves a significant reduction in the number of parameters and a significant improvement in the efficiency by substituting the fully connected layer with a sparse layer. Extensive experiments demonstrate that DSparsE consistently outperforms existing state-of-the-art methods.

**Strengths:**

1. The paper is clearly written and easy to follow.
2. This model combines dynamic layer and residual structure, wwhich enables neural networks to better perform information fusion

**Weaknesses:**

1.	The number of datasets is only 2, relatively limited.
2.	The supplementary experiments in this article are not sufficient. For instance, Figure 4 provides a comparison between DSparseE and one baseline on a single dataset. Both the number of baselines for comparison and the diversity of datasets should be increased. Additionally, Figure 5 only illustrates three scenarios of expert quantity, making it challenging to discern a clear and definitive trend. In Figure 6, the blue and orange lines are difficult to interpret. The author mentions that these are results with different numbers of residual layer depth but fails to provide specific numerical values for these quantities. The inadequacy of supplementary experiments has compromised the rigor and credibility of the results.
3.	In the "Contribution" section, the authors claim that their model achieves a significant reduction in the number of parameters and a significant improvement in model efficiency. However, the subsequent text does not provide a particularly clear substantiation of this claim. It would be beneficial to provide more explicit details, such as in terms of runtime or the exact number of parameters, to support this assertion.

**Questions:**

1.	Can you add more datasets?
2.	Can you improve the supplementary experiments?
3.	Can you provide more explicit details of the improvement of the efficiency?

---

> ### Author Response · Authors · 2023-11-19
>
> **We would like to sincerely thank you for the time and effort you have dedicated to carefully reading our manuscript and providing insightful comments and suggestions. We have thoroughly considered each point raised and have addressed them in the following sections, aiming to improve and refine our manuscript based on your valuable input.**
>
> ---
>
> >* **W1: The number of datasets is only 2, relatively limited.**
> >* **Q1: Can you add more datasets?**
>
> **Response**
> Thank you for your comment! We have added a new dataset, YAGO3-10[1], the details of which are as follows in the table below:
> | Dataset | Entity | Relation | Train | Valid | Test |
> | :---: | :---: | :---: |:---:|:---:|:---:|
> | YAGO3-10 | 123,182 | 37 | 1079,040 | 5000 | 5000 |
>
> Accordingly, we have conducted experiments on each metric and compared them with the baseline. Due to the fewer baselines of this dataset, we only selected models with data and presented the results in the table below.
>
> | Dataset | Hits@1 | Hits@10 | MRR |
> | :---: | :---: | :---: |:---:|
> | DistMult | 0.240 | 0.540 | 0.340 |
> | ConvE | 0.350 | 0.620 | 0.440 |
> | ComplEx | 0.260 | 0.550 | 0.360 |
> | RotatE | 0.402 | 0.670 | 0.495 |
> | InteractE | 0.462 | 0.687 | 0.541 |
> | **DSparsE** | **0.464** | **0.690** | **0.544** |
>
> Note that the result on this dataset may change in subsequent versions.

---

> > ### Author Response · Authors · 2023-11-19
> >
> > >* **W2: The supplementary experiments in this article are not sufficient. For instance, Figure 4 provides a comparison between DSparsE and one baseline on a single dataset. Both the number of baselines for comparison and the diversity of datasets should be increased. Additionally, Figure 5 only illustrates three scenarios of expert quantity, making it challenging to discern a clear and definitive trend. In Figure 6, the blue and orange lines are difficult to interpret. The author mentions that these are results with different numbers of residual layer depth but fails to provide specific numerical values for these quantities. The inadequacy of supplementary experiments has compromised the rigor and credibility of the results.**
> > >* **Q2: Can you improve the supplementary experiments?**
> >
> > **Response**
> >
> > Thank you for your comments. In the revised manuscript, we have conducted more experiments. Regarding the experimental section you mentioned, we have made the following improvements:
> >
> > For the original Figure 4, we added a comparison with InteractE, demonstrating the monotonic decline of InteractE with increasing sparsity. We have explained this in the revised version of the paper as follows:
> >
> > >On the other hand, the performance of the InteractE model demonstrates a consistent decrease with increasing levels of sparsity. This trend is attributed to the model architecture of InteractE, where only the final feature decoding layer is an MLP layer. Experimental results indicate that introducing increased sparsity over the sparse interactions already captured by the earlier convolutional layers adversely impacts the model's predictive performance.
> >
> > Another issue you raised was the insufficiency of datasets. Due to time and computational constraints, our experiments are still ongoing. However, for the WN18RR dataset, it demonstrates a pattern of performance change with sparsity similar to that of the FB15k-237 dataset. We can provide some of this data as follows:
> > | Sparsity | Hits@1 |
> > | :---: | :---: |
> > | 0. | 0.438 |
> > | 0.5 | 0.443 |
> > | 0.8 | 0.421 |
> >
> >
> > For the original Figure 5, we added data points for the number of expert blocks: 1, 50, 80. The graph clearly shows the trend of the prediction effect first increasing and then decreasing with the addition of more expert blocks. We now present this graph in a tabular form as follows:
> >
> > | Hits@1 | $n = 1$ | $n = 5$ | $n = 15$ | $n = 25$ |$n = 35$ | $n = 50$| $n = 80$ |
> > | :---: | :---: | :---: |:---: |:---:|:---:|:---:|:---: |
> > | $t = 10$ | .2647 | .2689 |.2681|.2692|.2678|.2654|.2647|
> > | $t = 20$ | .2647 | .2702 |.2716|.2674|.2680|.2659|.2649|
> > | $t = 30$ | .2647 | .2692 |.2699|.2673|.2671|.2651|.2644|
> > | Pure MLP | .2647 | .2652 |.2661 |.2648|.2639|.2671|.2605|
> >
> > In our revision we have given a detailed explanation for this result.
> >
> > >The experimental results indicate that the predictive performance initially increases and then decreases with the rising number of expert blocks. Initially, the addition of expert blocks can effectively enhance feature fusion capabilities. This can be explained from two perspectives. Firstly, in contrast to a non-partitioned fully connected structure (i.e., a very wide fully connected layer), the expert blocks in the dynamic layer represent a form of regular sparse connections (See Appendix D for details). These sparse connections are further integrated through a decision layer, namely a gating layer, forming a hyper-network structure, which brings robustness to the entire network. Secondly, the expert blocks in the dynamic layer can be viewed as sub-modules in an ensemble learning framework. This ensemble learning architecture can effectively suppress the propagation of errors, reducing the variance in prediction results. Under the hypernetwork's constraints, the multi-modular architecture can evolve towards the optimal direction (See Appendix E for details).
> >
> > >
> > >However, when the number of expert blocks becomes excessive, the performance deteriorates, which can be attributed to two factors. First, an increase in network parameters introduces additional training complexity, diminishing the network's generalization performance. Second, the gating network is fundamentally a multi-classifier. An excessive number of categories increases the decision-making complexity of the network, making it more prone to difficulties.
> >
> > >
> > >Another important influencing factor is the temperature of the dynamic layer. High temperature values make the weight combinations tend towards an average, leading to weight homogenization. Conversely, low temperature values can render many experts ineffective in learning, thus impacting the results.

---

> ### Author Response · Authors · 2023-11-19
>
> （Following the previous text）Regarding the original Figure 6, we apologize for the confusion caused by inaccurate markings. Our intention was to distinguish a sudden change in the horizontal axis values from 5 to 100 using a dashed line and a different color. However, this approach now seems redundant. In our revised paper, we have improved the expression of this figure and the interpretation of the results. Additionally, if you have doubts about whether the network should continue to deepen, you can refer to our response to reviewer jcNV's question 5, which we will include in our revised version.
>
> Furthermore, we have added some comparative experiments, such as ablation studies of dynamic layers and relation-aware layers, as well as comparisons of the effects of sparsity structure with dropout and downscaling of the network. Interestingly, by studying the hypernetwork formed by the gated layers' responses to different entity-relation pairs, we discovered that the hypernetwork can effectively model semantic information in the latent space. Along with our analysis, we also provided visualized results.
>
>
>
> ---
>
> >* **W3: In the "Contribution" section, the authors claim that their model achieves a significant reduction in the number of parameters and a significant improvement in model efficiency. However, the subsequent text does not provide a particularly clear substantiation of this claim. It would be beneficial to provide more explicit details, such as in terms of runtime or the exact number of parameters, to support this assertion.**
> >* **Q3: Can you provide more explicit details of the improvement of the efficiency?**
>
> **Response**
>
> Thank you for your comment. Indeed, increasing the number of expert blocks results in reduced parameter efficiency, and the fixed structure sparsification does not significantly reduce the number of parameters in the current computing architecture. This is because non-structured pruning does not alter the number of matrix multiplications, and the optimal sparsity of 0.5 mentioned in our paper is not high enough to utilize sparse matrix techniques for optimization. To avoid misleading information, we have removed the statements about parameter reduction from the contributions section of our paper.
>
> In the table below, we present the parameter count and corresponding runtime. Several points should be noted regarding these results:
>
> | Number of parameters | FB15k-237 | WN18RR |
> | :---: | :---: | :---: |
> | InteractE | 18M | 60M |
> | ComDensE | 66M | 33M |
> | DSparsE | 69M | 29M |
>
> |  model | Running time (per batch, $batch size = 128$) on FB15k-237 | on WN18RR |
> | :---: | :---: |:---:|
> | InteractE | 2.58s | 3.43s |
> | ComDensE | 2.61s | 3.60s |
> | DSparsE | 2.67s | 3.44s |
>
> 1. Increasing the number of expert cores does not significantly increase the number of parameters.
> 2. Relatively speaking, increasing the number of expert cores has a minimal impact on computational speed.
> 3. For hardware that supports unstructured pruning, introducing sparsity can save storage space and computational resources to a certain extent.
>
> ---
>
> **We would like to sincerely thank you for your valuable comments, and we hope that you are satisfied with our responses. We appreciate for your reconsideration of our revised manuscript. Thank you very much for your time and effort in reviewing our work!!**
>
> ---
> [1] Suchanek F M, Kasneci G, Weikum G. Yago: a core of semantic knowledge[C]//Proceedings of the 16th international conference on World Wide Web. 2007: 697-706.

---

> ### Author Response · Authors · 2023-11-21
> **We're grateful for your help in improving our work!**
>
> **We have attempted to address all the points you raised in your review. Could you please confirm whether our revised submission meets your expectations and resolves the issues you highlighted? We appreciate your guidance in this process.**

---

### Official Review · Reviewer_jcNV · 2023-10-31

**Soundness:** 3 good
**Presentation:** 2 fair
**Contribution:** 2 fair
**Rating:** 5
**Confidence:** 3

**Summary:**

This paper introduces a novel dynamic sparse embedding method DSparsE for knowledge graph completion task. The DSparsE is proposed to solve the drawbacks of prone to overfitting and constraints on network depth of ConDensE and the limitation in feature interaction and interpretability of InteractE. DSparsE includes three main modules, the dynamic MLP layer and the relation-aware MLP layer in the encoder, and residual blocks in the decoder, and it named as DSparsE because the sparse MLP is applied. DSparsE is evaluated on two common KGC benchmarks, FB15k-237 and WN18RR. The results show DSparsE is effective for KGC task.

**Strengths:**

1. Authors tried to investigate how powerful the MLP is for KGC task by developing model based on pure MLP layers. This is significantly different to existing methods and very interesting.
2. Though the experiment results of DSparsE is not comparable to state-of-the-art. But it shows DSparsE is effective for KGC tasks.

**Weaknesses:**

1. Though the overall model architecture is novel, the key advantage of pure MLP-based model such as DSparsE is unclear. As I understand, it is not efficiency since 15(35) experts are set for FB15k-237(WN18RR) with each expert represented by an MLP layer, which will introduce a lot extra parameters compared to 1 expert. It is also not superior performance, since the link prediction results of DSparsE is comparable to existing methods, such as RESCAL and ComDensE.
2. How do the different modules affect the performance of DSparsE is not well illustrated. For example, the output vectors from Dynamic MLP layer and the Sparse Relation-aware Layer are concatenated into one vector as the input of the decoder. It is unclear the output vector of which affects the results more significantly.
3. Some minor points:
* In Table 2, the best MRR result on FB15k-237 should be 0.396 from ConvKB.
* Both the blue line and orange dashed line represents marked as "with residual structure", it is a bit confusing.

**Questions:**

1. What is the key advantages of designing KGC models with pure MLP layers compared to the existing methods, such as tensor-decomposition models and translational models introduced in the related works?
2. The definition of deep learning models is unclear. Is the graph neural network model be deep learning models? Why are translation models not deep learning models?
3. The output vectors from Dynamic MLP layer and the Sparse Relation-aware Layer are concatenated into one vector as the input of the decoder. Which vector affect the results more?
4. What is the number of parameters of DSparsE model for FB15k-237 and WN18RR?  Is the number of parameters of DSparsE significantly more than baseline methods?
5. The results of 100 depth of the residual structures. Is this mean that 100 residual MLP blocks are used in the model? If yes, how long does it take to train the model?

---

> ### Author Response · Authors · 2023-11-19
>
> **Thank you very much for your insightful comments on our manuscript. We have addressed your comments and made revisions in our new manuscript. Below are detailed responses to your questions:**
>
> ---
> >* **W1: Though the overall model architecture is novel, the key advantage of pure MLP-based model such as DSparsE is unclear. As I understand, it is not efficiency since 15(35) experts are set for FB15k-237(WN18RR) with each expert represented by an MLP layer, which will introduce a lot extra parameters compared to 1 expert. It is also not superior performance, since the link prediction results of DSparsE is comparable to existing methods, such as RESCAL and ComDensE.**
>
> >* **Q1: What is the key advantages of designing KGC models with pure MLP layers compared to the existing methods, such as tensor-decomposition models and translational models introduced in the related works?**
>
>
>
> **Response**
>
> Thank you for your comments. The key advantages of DSparsE can be summarized in three points:
>
> 1. Dynamic Characteristics: The relation-aware layer in ComDensE[1] only focuses on the dynamic nature of network weights as relationships change, neglecting the dynamics associated with node changes. Indeed, designing a separate weight matrix for each node would lead to an explosive increase in parameters with the scale of the graph. Therefore, we adopt a hypernetwork structure to build a network where parameter weights can change with the input entity-relation pairs. This network can consider both entity and relation node information, and works well in conjunction with the relation-aware layer. Additional ablation studies in the revised version of the paper demonstrate that parallel dynamic and relation-aware layers can achieve better performance.
>
>     | Dynamic+Relation-aware | Dynamic layer only | Relation-aware layer only |
>     |:-------:|:-------:|:-------:|
>     | **0.272**| 0.252 | 0.264 |
>
>
> Furthermore, we have investigated how the hypernetwork affects the weights of the expert blocks. Here, we will quote a portion of our revised version:
> >Moreover, our investigation into the gating layer's outputs has unveiled some intriguing insights. Each entity-relation pair in the dataset, upon processing through the gating layer, yields an output vector $\bm{o}$. These high-dimensional vectors were subjected to tSNE reduction, with the resultant visualization displayed in Figure 8 and Figure 9. Each point in this figure represents a unique entity-relation pair, distinguished by varying colors corresponding to different relationships. The visualization result reveals the following observations:
> >1. A tendency for entity-relation pairs of the same relationship type to cluster together, indicating proximity within the output space of the gated layer outputs.
> >2. The spatial distribution of clusters is significantly influenced by the nature of the relationships. For instance, relationships denoting inverse meanings (e.g. nominee\_inv and nominee) or semantic opposites (e.g. place of birth vs. place of burial) exhibit a tendency to spatially diverge, exhibiting a unique central symmetry characteristic in the reduced dimensional space, Conversely, relationships with similar semantics (e.g. nationality and city town) are observed to be proximate in the latent space. This proved that DSparsE can capture various associations between entities and relations.
> >3. Alterations in the head entity of a relation pair result in minor shifts within the vector output, confined to a limited scope. Within a fixed relation, the relative positioning of nodes within its corresponding cluster does not display a discernible pattern. This phenomenon can be attributed to the relatively lower frequency of triples involving individual nodes compared to those associated with a particular relation type, posing challenges in accurately modeling semantic information for nodes. Still, certain examples, such as Mariah Carley and Dmitri Shostakovich—both notable in the music domain—demonstrate proximity within clusters pertaining to specific relations.

---

> > ### Author Response · Authors · 2023-11-19
> >
> > 2. Sparse Characteristics: Traditional neural networks face the issues of overfitting and generalization, which are more severe in knowledge graph link prediction models based on neural networks. This is due to the open-world assumption[2] in knowledge graphs, which means that the graphs are incomplete and the dataset knowledge is extremely limited compared to the actual distribution of knowledge. Thus, our model introduces a fixed sparse structure, which is different from the dropout feature determined at model initialization. Our inspiration comes from convolutional neural networks like ConvE[3] and InteractE[4], which have sparse connections with weight sharing. We consider random sparse connections between neurons without weight sharing and use a hyperparameter alpha to control the density of feature interactions. Further experiments also show that sparse connections on the basis of the existing dropout can optimize the network model's predictive performance. The results below show the effectiveness of sparse structure. Note that extra dropout represents setting dropout rate $\hat{p}$ $=$ $p + \alpha(1-p)$, where $\alpha$ denotes the sparsity degree and $p$ denotes the original dropout rate. Here we set $\alpha$ to 0.5. The results indicate that sparse structure can effectively enhance the performance.
> >
> > | model | Dropout + sparse | Extra dropout |
> > | :---: | :---: | :---: |
> > | DSparsE | **0.272** | 0.266 |
> > | ComDensE | 0.267 | 0.258 |
> >
> >
> > 3. Residual Characteristics: Existing research has shown that deepening network layers can enhance expressiveness[5], but this comes with convergence difficulty, which is a problem that also present in knowledge graph link prediction models. For example, the authors of ComDensE found that increasing the number of layers actually degrades network performance. We also found that simply adding front-end fully connected layers in matrix decomposition or similar models reduces their expressiveness. However, the only way for shallow models to maintain effectiveness with increasing graph scale is to increase the embedding dimension, leading to greater computational costs. If the embedding dimension is kept constant, performance declines with increasing graph scale (especially the number of entities), such as RESCAL's performance on WNRR18 being worse than on FB15k-237 with the same embedding dimension. In DSparsE, we applied expert layers to maximize feature extraction and fusion, requiring a powerful decoding module. In experiments, we found that 3-5 layers of residual neural networks provided good results on Fb15k-237 and YAGO3-10. To some extent, the introduction of residual layers allows the network to deepen further, maximizing its expressive potential. This is beneficial for large knowledge graphs like YAGO3-10.
> >
> > Although DSparsE's advantage over RESCAL on FB15k-237 is not apparent, it is demonstrated that DSparsE performs better than RESCAL on WN18RR. Although DistMult is simple, it performs poorly due to its lack of asymmetrical relationship modeling capabilities. Moreover, you mentioned comparisons with translation models like TransE[6], TransH[7], TransR[8], TransD[9]. To our knowledge, these models significantly lag behind ours on typical datasets (like FB15k, WNRR, YAGO series). Their inferior performance is due to their oversimplicity, making them unable to accurately model complex relationship types and capture semantic features. In contrast, DSparsE can model various semantic associations of entity-relation pairs, offering significant potential. Results on Hits@10 are given below:
> >
> > | model | FB15k-237 | WN18RR |
> > | :---: | :---: | :---: |
> > | TransE | 0.471 | 0.512 |
> > | TransH | 0.490 | 0.507 |
> > | TransD | 0.461 | 0.508 |
> > | TransR | 0.511 | 0.519 |
> > | RESCAL | 0.548 | 0.487 |
> > | **DSparsE** | **0.551** | **0.539** |
> >
> > We agree that increasing the size of expert blocks indeed increases the network's parameter count. However, the slightly increase of parameters is acceptable. Compared to ComDensE and InteractE, the number of parameters on FB15k-237 is slightly larger, while it is smaller on WN18RR. Results below also demonstrate that a slight increase in parameters does not significantly impact the running speed. On the other hand, we have confirmed that dynamic layers have a stable performance advantage over wide fully connected layers with the same number of parameters .
> >
> > | Number of parameters | FB15k-237 | WN18RR |
> > | :---: | :---: | :---: |
> > | InteractE | 18M | 60M |
> > | ComDensE | 66M | 33M |
> > | DSparsE | 69M | 29M |
> >
> > |  | Running time (per batch, $batch size = 128$) on FB15k-237 | on WN18RR |
> > | :---: | :---: |:---:|
> > | InteractE | 2.58s | 3.43s |
> > | ComDensE | 2.61s | 3.60s |
> > | DSparsE | 2.67s | 3.44s |
> >
> > | model | Dynamic layer | MLP layer with same parameter count |
> > | :---: | :---: | :---: |
> > | **DSparsE** | **0.272** | 0.266 |

---

> > > ### Author Response · Authors · 2023-11-19
> > >
> > > >* **W2: How do the different modules affect the performance of DSparsE is not well illustrated. For example, the output vectors from Dynamic MLP layer and the Sparse Relation-aware Layer are concatenated into one vector as the input of the decoder. It is unclear the output vector of which affects the results more significantly.**
> > > >* **Q3: The output vectors from Dynamic MLP layer and the Sparse Relation-aware Layer are concatenated into one vector as the input of the decoder. Which vector affect the results more?**
> > >
> > > **Response**
> > >
> > > Thank you for your comments. In the revised manuscript, we have supplemented more  experiments to illustrate the role of different parts. In addition, we have also supplemented more experiments to demonstrate the impact of the number of expert blocks on the final results, as well as the effects of replacing the sparse structure with additional dropout or reducing the number of neurons, as shown below:
> > >
> > > | Hits@1 | $n = 1$ | $n = 5$ | $n = 15$ | $n = 25$ |$n = 35$ | $n = 50$| $n = 80$ |
> > > | :---: | :---: | :---: |:---: |:---:|:---:|:---:|:---: |
> > > | $t = 10$ | .2647 | .2689 |.2681|.2692|.2678|.2654|.2647|
> > > | $t = 20$ | .2647 | .2702 |.2716|.2674|.2680|.2659|.2649|
> > > | $t = 30$ | .2647 | .2692 |.2699|.2673|.2671|.2651|.2644|
> > > | Pure MLP | .2647 | .2652 |.2661 |.2648|.2639|.2671|.2605|
> > >
> > > In our revision, we have also given a detailed explanation for this result.
> > >
> > > >The experimental results indicate that the predictive performance initially increases and then decreases with the rising number of expert blocks. Initially, the addition of expert blocks can effectively enhance feature fusion capabilities. This can be explained from two perspectives. Firstly, in contrast to a non-partitioned fully connected structure (i.e., a very wide fully connected layer), the expert blocks in the dynamic layer represent a form of regular sparse connections (See Appendix D for details). These sparse connections are further integrated through a decision layer, namely a gating layer, forming a hyper-network structure, which brings robustness to the entire network[10]. Secondly, the expert blocks in the dynamic layer can be viewed as sub-modules in an ensemble learning framework. This ensemble learning architecture can effectively suppress the propagation of errors, reducing the variance in prediction results. Under the hypernetwork's constraints, the multi-modular architecture can evolve towards the optimal direction (See Appendix E for details).
> > > >
> > > >However, when the number of expert blocks becomes excessive, the performance deteriorates, which can be attributed to two factors. First, an increase in network parameters introduces additional training complexity, diminishing the network's generalization performance. Second, the gating network is fundamentally a multi-classifier. An excessive number of categories increases the decision-making complexity of the network, making it more prone to difficulties.
> > > >
> > > >Another important influencing factor is the temperature of the dynamic layer. High temperature values make the weight combinations tend towards an average, leading to weight homogenization. Conversely, low temperature values can render many experts ineffective in learning, thus impacting the results.
> > >
> > > | model | Dropout + sparse | Extra dropout | Downscale |
> > > | :---: | :---: | :---: |:---:|
> > > | **DSparsE** | **0.272** | 0.266 | 0.265 |
> > > | ComDensE | 0.265 | 0.258 | 0.263 |
> > >
> > > >To demonstrate the effectiveness of sparse structure, we have implemented two modifications to the network. Firstly, we  reduce the output dimension in the linear layers to $\alpha$ times their original number, ensuring the same level of interaction. Specifically, for a linear layer with output dimension $d$, we set the output dimension to $\hat{d} = \alpha d$. Secondly, we actively increase the dropout rate to $\hat{p} = p + \alpha(1 - p)$, where $p$ is the original dropout rate. The results are shown above. It indicates that actively decreasing the number of neurons significantly reduced performance, whereas actively increasing dropout rate drastically deteriorated the final outcomes. This is due to the fact that reducing the neuron number confines the output to a smaller subspace, limiting expressive freedom. Simultaneously, since each training iteration changes the dropout mask, an excessively high dropout actively introduces more uncertainty, thus diminishing network stability.

---

> > > > ### Author Response · Authors · 2023-11-19
> > > >
> > > > >* **W3: Some minor points:**
> > > > **In Table 2, the best MRR result on FB15k-237 should be 0.396 from ConvKB. Both the blue line and orange dashed line represents marked as "with residual structure", it is a bit confusing.**
> > > >
> > > > **Response**
> > > > Oh! What a silly mistake! We apologize for any inconvenience caused by our oversight. Indeed, the original version of our paper contained some basic issues, such as problems with the labeling of figures and data. In the revision, we have thoroughly revamped the article, fixing the original issues. Allow me to introduce the main changes we have made:
> > > >
> > > > 1. We have polished and updated the language of the article, especially the explanations in the abstract and the contributions section.
> > > > 2. Following the suggestions of other reviewers, we have modified and added several necessary charts to aid understanding and enriched the captions for these charts.
> > > > 3. In line with comments from other reviewers, we have added the YAGO3-10 dataset and provided corresponding baseline data, demonstrating that our model outperforms all similar models.
> > > > 4. Following your advice, we added multiple comparative experiments and ablation studies, elucidating the mechanisms of each part.
> > > > 5. We have included additional data on the number of parameters and runtime.
> > > > 6. Our study of the latent space output has yielded interesting findings, which were mentioned in response to the first question. However, we hope that you will review them in conjunction with the revised version's images.
> > > > 7. Due to space constraints, we have placed some additional content in an appendix to aid understanding. Except for the parts related to the number of parameters, all other sections are referenced in the main text.
> > > >
> > > > We sincerely invite you to revisit and review our revised manuscript. Your insights and comments are invaluable to us, and we have worked diligently to address the issues and enhance the quality of our work. We look forward to your feedback on these improvements!
> > > >
> > > > ---
> > > > >* **Q2: The definition of deep learning models is unclear. Is the graph neural network model be deep learning models? Why are translation models not deep learning models?**
> > > >
> > > > **Response**
> > > > Thank you for your comment. Deep learning models are typically considered to be hierarchical assemblies of multiple layers of neural networks. Graph Neural Networks (GNNs) also fall under deep learning models, as their information aggregation process usually consists of multiple layers of graph convolution. As you and another reviewer pointed out, GNNs are widely applied in link prediction tasks in knowledge graphs, effectively capturing the contextual information of graph structures. Following your suggestions, we have supplemented the introduction of GNNs in the deep learning models section of our related work. However, contrary to intuition, some existing baselines such as CompGCN[11] and R-GCN[12] do not perform as well as expected. Additionally, the diversity of graph structures poses a challenge to parallel training, drastically reducing training efficiency. Of course, we are eager to actively explore more potentialities of GNNs in our future work.
> > > >
> > > > Nevertheless, translation models like TransE and TransD are generally not considered deep models. Firstly, the Trans series models are very shallow and lack the structure of traditional neural networks. These models perform gradient descent training on graph embeddings through a designed scoring function. Although there are some linear transformation matrices that need to be learned in TransD and TransR, they lack the basic nonlinear layers found in neural networks, limiting the expressive power of these types of models.
> > > >
> > > > Of course, if you have concrete evidence to prove that our understanding is incorrect, please point it out, and we are very willing to make the relevant adjustments in our paper.
> > > >
> > > > >* **Q4: What is the number of parameters of DSparsE model for FB15k-237 and WN18RR? Is the number of parameters of DSparsE significantly more than baseline methods?**
> > > >
> > > > **Response**
> > > > This is indeed a key concern. Please refer to the answer for W1 and Q1 for detailed information.

---

> ### Author Response · Authors · 2023-11-19
>
> >* **Q5: The results of 100 depth of the residual structures. Is this mean that 100 residual MLP blocks are used in the model? If yes, how long does it take to train the model?**
>
> **Response**
> We apologize for any confusion caused by our unclear description. It should be clarified that for a small dataset like FB15k-237, a massive network with 100 layers is extremely redundant. For this dataset, the optimal depth of the decoding layers is around 3 layers.
>
> We fully understand your concerns. Your core question is:
> * **_Is it really necessary to deepen the network?_**
>
> In fact, we have observed that the larger the dataset, the deeper the required depth of the decoding layer. Blow is the relationship between the scale of the dataset and the corresponding optimal number of layers:
>
> | Dataset | Entity | Relation | Train | optimal residual layer depth |
> | :---: | :---: | :---: |:---:|:---:|
> | FB15k-237 | 14,541 | 237 | 272,115 | 3 |
> | WN18RR | 40,943 | 11 | 86,835 | 1 |
> | YAGO3-10 | 123,182 | 37 | 1079,040 | 5 |
>
>
> Based on this fact, we make a reasonable conjecture that for super-large-scale knowledge graphs (like FreeBase, DBPedia, etc.) with tens of billions of knowledge triples, the required number of decoding layers will increase further. Under this assumption, our experiments prove that incorporating residual connections in MLPs can effectively maintain the network's decoding performance and maximize the network's expressive power.
>
> You also mentioned the issue of training time for deep networks. We have conducted corresponding experiments, and the results are shown in the following table. Interestingly, stacking a large number of linear layers does not significantly increase the network's computational time. This may be because computational resources are mainly consumed during preprocessing and information transmission stages. To ensure rigor, we have specified the models of the GPUs and CPUs used for reference.
>
> | CPU | GPU | Running time for $depth = 3$ | running time for $depth = 100$ |
> | :---: | :---: |:---:|:---:|
> | AMD Ryzen 9 7945HX | NVIDIA GeForce RTX 4080 Laptop GPU | 10.2s per batch(512) | 13.7s per batch(512) |
>
>
> ---
> **Again, we respectfully request your review of the latest revision. We are eager to hear your comments on our revised manuscript to further improve our work.**
>
> **Thank you once again for your valuable comments on our research. It is truly rewarding that you find our work to be _very interesting_! Wish you have a nice day! :D**
>
> ---
>
> [1] Kim M, Baek S. Comdense: combined dense embedding of relation-aware and common features for knowledge graph completion[C]//2022 26th International Conference on Pattern Recognition (ICPR). IEEE, 2022: 1989-1995.
>
> [2] Drummond N, Shearer R. The open world assumption[C]//eSI Workshop: The Closed World of Databases meets the Open World of the Semantic Web. 2006, 15: 1.
>
> [3] Dettmers T, Minervini P, Stenetorp P, et al. Convolutional 2d knowledge graph embeddings[C]//Proceedings of the AAAI conference on artificial intelligence. 2018, 32(1).
>
> [4] Vashishth S, Sanyal S, Nitin V, et al. Interacte: Improving convolution-based knowledge graph embeddings by increasing feature interactions[C]//Proceedings of the AAAI conference on artificial intelligence. 2020, 34(03): 3009-3016.
>
> [5] Chatterjee S, Zielinski P. On the generalization mystery in deep learning[J]. arXiv preprint arXiv:2203.10036, 2022.
>
> [6] Bordes A, Usunier N, Garcia-Duran A, et al. Translating embeddings for modeling multi-relational data[J]. Advances in neural information processing systems, 2013, 26.
>
> [7] Wang Z, Zhang J, Feng J, et al. Knowledge graph embedding by translating on hyperplanes[C]//Proceedings of the AAAI conference on artificial intelligence. 2014, 28(1).
>
> [8] Lin Y, Liu Z, Sun M, et al. Learning entity and relation embeddings for knowledge graph completion[C]//Proceedings of the AAAI conference on artificial intelligence. 2015, 29(1).
>
> [9] Ji G, He S, Xu L, et al. Knowledge graph embedding via dynamic mapping matrix[C]//Proceedings of the 53rd annual meeting of the association for computational linguistics and the 7th international joint conference on natural language processing (volume 1: Long papers). 2015: 687-696.
>
> [10] Jacobs R A, Jordan M I, Nowlan S J, et al. ªAdaptive Mixtures of Local Experts, º Neural Computation, vol. 3[J]. 1991.
>
> [11] Vashishth S, Sanyal S, Nitin V, et al. Composition-based multi-relational graph convolutional networks[J]. arXiv preprint arXiv:1911.03082, 2019.
>
> [12] Schlichtkrull M, Kipf T N, Bloem P, et al. Modeling relational data with graph convolutional networks[C]//The Semantic Web: 15th International Conference, ESWC 2018, Heraklion, Crete, Greece, June 3–7, 2018, Proceedings 15. Springer International Publishing, 2018: 593-607.

---

> > ### Author Response · Authors · 2023-11-21
> > **Please let us know if there are any further questions :D**
> >
> > **Following your insightful remarks, we have amended our manuscript. May we kindly inquire if our revisions have fully addressed your concerns? Your expert opinion is crucial to the enhancement of our work.**

---

> > > ### Comment · Reviewer_jcNV · 2023-11-22
> > > **Thank the authors for response**
> > >
> > > Thank the authors for their detailed response. I would like to keep my initial score.

---

### Official Review · Reviewer_5L59 · 2023-11-01

**Soundness:** 3 good
**Presentation:** 2 fair
**Contribution:** 2 fair
**Rating:** 3
**Confidence:** 4

**Summary:**

This paper proposed a new architecture of neural networks, the DSparsE, for graph completion. It is a link prediction model structure that uses only MLP layers and employs sparse and residual structures to alleviate overfitting, and reduce the difficulty of training deep networks. The paper provides performance comparison of various knolwege graph embedding techniques across two datasets.

**Strengths:**

- The proposed architecture utilizes various methods to prevent some of the well known problems, especially the overfitting problem which is important for knowledge graph completion.
- The paper provides ablation studies to show the benefit of implementing proposed methods.

**Weaknesses:**

- Better explanations can be included. For instance, the paper states the expert kernels, but does not explicitly explain what is the expert kernels or why it is "expert".
- Better figures and figure captions can be written. The architecture figures are small and the explanations in the captions is very limited.
- Some statistical testings (or critical plots) for the comparing methods would make the arguments of the paper stronger.
- Comparison of computational time would be necessary to address the computational complexity problems in the existing methods.

**Questions:**

- What is an expert kernel?
- How does the model do in the case of no experts? Or simple mean of the outputs of k different mlps.
- How is imposing sparse structure different from the drop-connect (dropout on the weights)
- Why does the dynamic structure enhance the robustness of the model? Is it by some form of ensembling?
- How statistically different are the results between the models (or at least the top competing ones).
- How does the model perform in terms of computational complexity?

---

> ### Author Response · Authors · 2023-11-19
>
> **We would like to sincerely thank you for the time and effort you have dedicated to carefully reading our manuscript and providing insightful comments and suggestions. Here is our response.**
>
> ---
>
> >* **W1: Better explanations can be included. For instance, the paper states the expert kernels, but does not explicitly explain what is the expert kernels or why it is "expert".**
> >* **Q1：What is an expert kernel?**
> >* **Q2: How does the model do in the case of no experts? Or simple mean of the outputs of k different mlps.**
> >* **Q4: Why does the dynamic structure enhance the robustness of the model? Is it by some form of ensembling?**
>
> **Response**
>
>
> Thank you for your comment. As per your suggestion regarding the term “expert kernel,” we have revised it to “expert block” for clarity and added explanatory sentences. We have also included a figure specifically for the dynamic layer section for better understanding. The revised statements concerning the experts are as follows:
>
> >The encoding part consists of an MLP layer with $k$ expert blocks (i.e., structurally similar MLP sub-blocks) and a relation-aware MLP layer.
>
> For an explanation of the term _expert_, you can refer to the paper _Adaptive mixtures of local experts_[1] as well as _Outrageously large neural networks: The sparsely-gated mixture-of-experts layer_[2]. The explanation provided therein is as follows:
> >_We present a new supervised learning procedure for systems composed of many separate networks, each of which learns to handle a subset of the complete set of training cases. The new procedure can be viewed either as a modular version of a multilayer supervised network, or as an associative version of competitive learning[1]._
>
> > _The experts are themselves neural networks, each with their own parameters. Although in principle we only require that the experts accept the same sized inputs and produce the same-sized outputs, in our initial investigations in this paper, we restrict ourselves to the case where the models are feed-forward networks with identical architectures, but with separate parameters[2]._
>
> From the explanation, it becomes evident that what is referred to as an _expert_ is essentially a sub-neural network with similar (or the same) structure. Training a large number of such neural networks in parallel can enhance the model's robustness. This aspect has also been elaborated in the experimental section of the revised paper (which includes comparisons with a pure MLP layer without dynamic adjustments). Moreover, reviewer S6XC mentioned in W2 that the original Figure 5 lacked sufficient experimental data on the number of expert blocks. We have now increased the number of data points, which reveals a rather distinct trend. The revised elucidation for this part of the experiment is as follows:
>
> >The experimental results indicate that the predictive performance first increases and then decreases with the rising number of expert blocks. Initially, the addition of expert blocks can effectively enhance feature fusion capabilities. This can be explained from two perspectives. Firstly, in contrast to a non-partitioned fully connected structure (i.e., a very wide fully connected layer), the expert blocks in the dynamic layer represent a form of regular sparse connections (See Appendix D for details). These sparse connections are further integrated through a decision layer, namely a gating layer, forming a hyper-network structure, which brings robustness to the entire network. Secondly, the expert blocks in the dynamic layer can be viewed as sub-modules in an ensemble learning framework. This ensemble learning architecture can effectively suppress the propagation of errors, reducing the variance in prediction results. Under the hypernetwork's constraints, the multi-modular architecture can evolve towards the optimal direction (See Appendix E for details).
>
> >
> >However, when the number of expert blocks becomes excessive, the performance deteriorates, which can be attributed to two factors. First, an increase in network parameters introduces additional training complexity, diminishing the network's generalization performance. Second, the gating network is fundamentally a multi-classifier. An excessive number of categories increases the decision-making complexity of the network, making it more prone to difficulties.
>
> >
> >Another important influencing factor is the temperature of the dynamic layer. High temperature values make the weight combinations tend towards an average, leading to weight homogenization. Conversely, low temperature values can render many experts ineffective in learning, thus impacting the results.

---

> ### Author Response · Authors · 2023-11-19
>
> (Following the previous text) In addition, we further investigate how the hypernetwork affects the weights of the expert blocks. In this regard, we have made some interesting observations. Here we will quote a part of the modified version:
>
> >Moreover, our investigation into the gating layer's outputs has unveiled some intriguing insights. Each entity-relation pair in the dataset, upon processing through the gating layer, yields an output vector $\bm{o}$. These high-dimensional vectors were subjected to tSNE reduction, with the resultant visualization displayed in Figure 8 and Figure 9. Each point in this figure represents a unique entity-relation pair, distinguished by varying colors corresponding to different relationships. The visualization result reveals the following observations:
> >1. A tendency for entity-relation pairs of the same relationship type to cluster together, indicating proximity within the output space of the gated layer outputs.
> >2. The spatial distribution of clusters is significantly influenced by the nature of the relationships. For instance, relationships denoting inverse meanings (e.g. nominee\_inv and nominee) or semantic opposites (e.g. place of birth vs. place of burial) exhibit a tendency to spatially diverge, exhibiting a unique central symmetry characteristic in the reduced dimensional space, Conversely, relationships with similar semantics (e.g. nationality and city town) are observed to be proximate in the latent space. This proved that DSparsE can capture various associations between entities and relations.
> >3. Alterations in the head entity of a relation pair result in minor shifts within the vector output, confined to a limited scope. Within a fixed relation, the relative positioning of nodes within its corresponding cluster does not display a discernible pattern. This phenomenon can be attributed to the relatively lower frequency of triples involving individual nodes compared to those associated with a particular relation type, posing challenges in accurately modeling semantic information for nodes. Still, certain examples, such as Mariah Carley and Dmitri Shostakovich—both notable in the music domain—demonstrate proximity within clusters pertaining to specific relations.
>
> For the images mentioned in the text, please refer to our revised version of the paper.
>
> ---
>
> >* **W2: Better figures and figure captions can be written. The architecture figures are small and the explanations in the captions is very limited.**
>
> **Response**
> Thank you for your comment. we have added detailed captions to all images and provided thorough descriptions within the text in the revised version. Regarding the figures, we have made the revisions as follows:
>
> 1. We enlarged the overall architecture diagram and bolded the key text within it. If you still see any issues, please do not hesitate to point them out, and we will continue to modify it to meet your requirements.
> 2. We updated the architecture diagram of the residual layer (changed from horizontal to vertical orientation) and added a dynamic layer architecture diagram to aid understanding.
> 3. We moved the original image for intuitive understanding of the sparse layer to the appendix due to space constraints.
>
> In addition, we have completely revamped the structure of the paper and optimized the language expression in key sections. Specifically, some of the main changes are as follows:
>
>
> 1. We have polished and updated the language of the article, especially the explanations in the abstract and the contributions section.
> 2. We have modified and added several necessary charts to aid understanding and enriched the captions for these charts.
> 3. We have added the YAGO3-10 dataset and provided corresponding baseline data, demonstrating that our model outperforms all similar models.
> 4.We have added multiple comparative experiments and ablation studies, elucidating the mechanisms of each part.
> 5. We have included additional data on the number of parameters and runtime.
> 6. Our study of the latent space output has yielded interesting findings, which were mentioned in response to the first question. However, we hope that you will review them in conjunction with the revised version's images.
> 7. Due to space constraints, we have placed some additional content in an appendix to aid understanding. Except for the parts related to the number of parameters, all other sections are referenced in the main text.

---

> ### Author Response · Authors · 2023-11-19
>
> ---
>
> >* **W3: Some statistical testings (or critical plots) for the comparing methods would make the arguments of the paper stronger.**
> >* **Q3: How is imposing sparse structure different from the drop-connect (dropout on the weights)**
> >* **Q5: How statistically different are the results between the models (or at least the top competing ones).**
>
> **Response**
> Thank you for your constructive suggestions. In the revised version of the paper, we have added a dataset, YAGO3-10. Now we present the experimental results of DSparsE on the YAGO3-10 dataset across various metrics:
>
> | Dataset | Entity | Relation | Train | Valid | Test |
> | :---: | :---: | :---: |:---:|:---:|:---:|
> | YAGO3-10 | 123,182 | 37 | 1079,040 | 5000 | 5000 |
>
>
> | Dataset | Hits@1 | Hits@10 | MRR |
> | :---: | :---: | :---: |:---:|
> | DistMult | 0.240 | 0.540 | 0.340 |
> | ConvE | 0.350 | 0.620 | 0.440 |
> | ComplEx | 0.260 | 0.550 | 0.360 |
> | RotatE | 0.402 | 0.670 | 0.495 |
> | InteractE | 0.462 | 0.687 | 0.541 |
> | **DSparsE** | **0.464** | **0.690** | **0.544** |
>
> We also give the detailed results as well as the comparison on FB15k-237:
>
> |                    |      | MRR   | Hits@10 | Hits@1 | MRR   | Hits@10 | Hits@1 | MRR   | Hits@10 | Hits@1 |
> |--------------------|------|-------|---------|--------|-------|---------|--------|-------|---------|--------|
> |                    | InteractE  |  |    | ComDensE  |  |    |   | DSparsE |   | |
> | **Pred Head**      | 1:1  | 0.386 | 0.547   | 0.245  | 0.422 | 0.557   | 0.349  | **0.434** | **0.572**   | **0.358**|
> |                    | 1:N  | **0.106** | **0.192**   | 0.043  | 0.084 | 0.181   | 0.043  | 0.101 | 0.185   | **0.044**|
> |                    | N:1  | 0.466 | 0.647   | 0.369  | 0.466 | 0.649   | 0.372  | **0.467** | **0.655**   | **0.376**|
> |                    | N:N  | 0.276 | 0.476   | 0.164  | 0.279 | 0.476   | 0.187  | **0.287** | **0.494**   | **0.195**|
> | **Pred Tail**      | 1:1  | 0.368 | 0.547   | 0.229  | 0.422 | 0.563   | 0.349  | **0.428** | **0.570**   | **0.351**|
> |                    | 1:N  | 0.777 | 0.708   | **0.881** | **0.779** | 0.884   | 0.717  | 0.778 | **0.886**   | 0.796|
> |                    | N:1  | 0.074 | 0.141   | 0.034  | 0.084 | 0.169   | **0.043** | **0.088** | **0.171**   | 0.042|
> |                    | N:N  | 0.395 | 0.617   | 0.272  | **0.396** | 0.618   | 0.285  | 0.395 | **0.624**   | **0.286**|
>
> For the original Figure 4, we added a comparison with InteractE, demonstrating the monotonic decline of InteractE with increasing sparsity. We have explained this in the revised version of the paper as follows:
>
> >On the other hand, the performance of the InteractE model demonstrates a consistent decrease with increasing levels of sparsity. This trend is attributed to the model architecture of InteractE, where only the final feature decoding layer is an MLP layer. Experimental results indicate that introducing increased sparsity over the sparse interactions already captured by the earlier convolutional layers adversely impacts the model's predictive performance.
>
> While investigating the impact of sparsity, we also conducted partial tests of DSparsE on the WN18RR dataset. The experiments are still ongoing, but we provide some of the data as follows, which is consistent with the analytical conclusions in the original text:
>
> | Sparsity | Hits@1 |
> | :---: | :---: |
> | 0. | 0.438 |
> | 0.3 | 0.441 |
> | 0.5 | 0.443 |
> | 0.8 | 0.421 |
>
> Additional ablation studies in the revised version of the paper prove that parallel dynamic and relation-aware layers can achieve better performance.
>
> | Dynamic+Relation-aware | Dynamic layer only | Relation-aware layer only |
> |:-------:|:-------:|:-------:|
> | **0.272**| 0.252 | 0.264 |
>
>
>
> The Dropout method applies different masks with each training iteration, which adds to the instability of the training. In our method, the pattern of sparse connections is already determined at initialization. Essentially, our method cuts the connections between neurons, weakening the intensity of feature interactions. through empirical evidence, it was demonstrated that:
>
> 1. Our method has advantages over simply reducing the network size (i.e., the number of output neurons).
> 2. Our method has advantages over the use of Dropout alone.

---

> > ### Author Response · Authors · 2023-11-19
> >
> > (Following the previous text)So we explore the effects of replacing the sparse structure with additional dropout or reducing the number of neurons, as shown below:
> >
> > | model | Dropout + sparse | Extra dropout | Downscale |
> > | :---: | :---: | :---: |:---:|
> > | **DSparsE** | **0.272** | 0.266 | 0.265 |
> > | ComDensE | 0.265 | 0.258 | 0.263 |
> >
> > >The results demonstrate that enhancing a network's effectiveness can be achieved by introducing random sparsity. However, this raises two questions:
> >     > 1. __Can actively reducing the scale of the linear layer yield a similar effect?__
> >     > 2. __Can a similar outcome be achieved by actively increasing the Dropout probability?__
> > >
> > >To demonstrate the effectiveness of sparse structure, we actively implemented two modifications to the network. Firstly, we actively reduce the output dimension in the linear layers to $\alpha$ times their original number, ensuring the same level of interaction. Specifically, for a linear layer with output dimension $d$, we set the output dimension to $\hat{d} = \alpha d$. Secondly, we actively increase the dropout rate to $\hat{p} = p + \alpha(1 - p)$, where $p$ is the original dropout rate. The result are shown above. It indicates that actively decreasing the number of neurons significantly reduced performance, whereas actively increasing dropout rate drastically deteriorated the final outcomes. This is due to the fact that reducing the neuron number confines the output to a smaller subspace, limiting expressive freedom. Simultaneously, since each training iteration changes the dropout mask, an excessively high dropout actively introduces more uncertainty, thus diminishing network stability.
> >
> > Furthermore, as we described in our response to Q1, we have studied the characteristics of the data distribution in the latent subspace output by the gating layer and have provided interesting visual results.
> >
> > In addition, we have included some data and charts in the appendix of the revised paper, which will help readers better understand the principles and effects of our model.
> >
> > ---
> > >* **W4: Comparison of computational time would be necessary to address the computational complexity problems in the existing methods.**
> > >* **Q6: How does the model perform in terms of computational complexity?**
> >
> > **Response**
> >
> > In the table below, we present the parameter count and corresponding runtime. Several points should be noted regarding these results:
> >
> > | Number of parameters | FB15k-237 | WN18RR |
> > | :---: | :---: | :---: |
> > | InteractE | 18M | 60M |
> > | ComDensE | 66M | 33M |
> > | DSparsE | 69M | 29M |
> >
> > |  | Running time (per batch, $batch size = 128$) on FB15k-237 | on WN18RR |
> > | :---: | :---: |:---:|
> > | InteractE | 2.58s | 3.43s |
> > | ComDensE | 2.61s | 3.60s |
> > | DSparsE | 2.67s | 3.44s |
> >
> >
> >
> >
> > The statistical results of the parameter count show that DSparsE has a similar number of parameters on FB15k-237 compared to ComDensE, and is slightly lower on WN18RR than ComDensE and InteractE.
> >
> > We acknowledge that this parameter count does not offer a clear advantage, but at the same time, we offer the following explanation:
> >
> > 1. Increasing the number of expert cores does not significantly increase the parameter count.
> > 2. Relatively speaking, increasing the number of expert cores has a minimal impact on computational speed.
> > 3. For hardware that supports unstructured pruning, introducing sparsity can save storage space and computational resources to a certain extent.
> >
> > For other models with fewer parameters, such as improved versions of RESCAL[3] and the Trans series models[4, 5, 6, 7], although they have fewer parameters, their prediction performance is far from our model. Below are some of the results:
> >
> > | model | FB15k-237 | WN18RR |
> > | :---: | :---: | :---: |
> > | TransE | 0.471 | 0.512 |
> > | TransH | 0.490 | 0.507 |
> > | TransD | 0.461 | 0.508 |
> > | TransR | 0.511 | 0.519 |
> > | RESCAL | 0.548 | 0.487 |
> > | **DSparsE** | **0.551** | **0.539** |
> >
> > ---
> > **We appreciate the careful consideration you have given to our manuscript. In response, we have thoroughly revised the manuscript and hope that you will be able to take the time to read it again.**
> >
> > **Once again, we express our heartfelt gratitude for your time and input. We are looking forward to your further consideration.**

---

> > > ### Author Response · Authors · 2023-11-19
> > >
> > > [1] Jacobs R A, Jordan M I, Nowlan S J, et al. Adaptive mixtures of local experts[J]. Neural computation, 1991, 3(1): 79-87.
> > >
> > > [2] Shazeer N, Mirhoseini A, Maziarz K, et al. Outrageously large neural networks: The sparsely-gated mixture-of-experts layer[J]. arXiv preprint arXiv:1701.06538, 2017.
> > >
> > > [3] Kong X, Chen X, Hovy E. Decompressing knowledge graph representations for link prediction[J]. arXiv preprint arXiv:1911.04053, 2019.
> > >
> > > [4] Bordes A, Usunier N, Garcia-Duran A, et al. Translating embeddings for modeling multi-relational data[J]. Advances in neural information processing systems, 2013, 26.
> > >
> > > [5] Wang Z, Zhang J, Feng J, et al. Knowledge graph embedding by translating on hyperplanes[C]//Proceedings of the AAAI conference on artificial intelligence. 2014, 28(1).
> > >
> > > [6] Lin Y, Liu Z, Sun M, et al. Learning entity and relation embeddings for knowledge graph completion[C]//Proceedings of the AAAI conference on artificial intelligence. 2015, 29(1).
> > >
> > > [7] Ji G, He S, Xu L, et al. Knowledge graph embedding via dynamic mapping matrix[C]//Proceedings of the 53rd annual meeting of the association for computational linguistics and the 7th international joint conference on natural language processing (volume 1: Long papers). 2015: 687-696.

---

> ### Author Response · Authors · 2023-11-21
> **Did we fully address your concerns?**
>
> **We appreciate your insightful comments and suggestions. Could you kindly confirm if our responses have adequately addressed your concerns and queries? We aim to ensure clarity and completeness in our revisions. Your feedback is highly valuable to us.**

---

### Author Response · Authors · 2023-11-20
**Please give us a reply!**

**Dear Reviewers,**

**We are pleased to inform you that we have thoroughly revised our paper. The changes are mainly focused on the following points:**

**1. We have revamped the structure of the paper.**

**2. We have updated the contribution and abstract, and added many new experiments and discussions of the results.**

**3. We have added a new dataset, YAGO3-10.**

**4. We have polished the language and expression of the figures in the paper. Following the reviewers' suggestions, we have added detailed captions to all images and charts.**

**5. We have enriched the content of the appendix to aid understanding, especially adding a section on parameter efficiency.**

**On the basis of the revised version, we have made detailed and targeted responses to the comments of each reviewer. We sincerely request all reviewers to reconsider our revised version and we look forward to further suggestions from all the reviewers.**

**Again, we kindly request you to review the revised version before the rebuttal period ends. We believe that our responses and the changes made in the manuscript will address your concerns. We look forward to your further suggestions.**

---

### Meta-Review · Area_Chair_jrGk · 2023-12-09

**Metareview:**

This paper proposes a knowledge graph link prediction model, DSparsE, for graph completion.
DSparsE includes three main modules, the dynamic MLP layer and the relation-aware MLP layer in the encoder, and residual blocks in the decoder. It alleviates overfitting, and reduces the difficulty of training deep networks. Experimental results demonstrate that DSparsE outperforms several state-of-the-art methods on several datasets.

Firstly, the proposed method uses only MLP layers, which is different from existing methods. Secondly, the paper tries to prevent overfitting, which is important for knowledge graph completion. Finally, ablation studies are provided to show the effectiveness of the method.

However, the novelty of the method is limited, and the efficiency is not very obvious. Secondly, more datasets should be added, and more ablation studies should be provided, such as providing standard deviations. Finally, the paper should be further polished carefully, such as improve the figures.

**Justification For Why Not Higher Score:**

The novelty of the method is limited, and the efficiency is not very obvious. More experiments should be provided.

**Justification For Why Not Lower Score:**

N/A

---

### Decision · Program_Chairs · 2024-01-16

Reject